# From Spectrum-free towards Baseline-view-free: Double-track Proximity Driven Multi-view Clustering

**Shengju Yu** [1]   **Zhibin Dong** [1]   **Siwei Wang** [2]   **Suyuan Liu** [1]   **Ke Liang** [1]   **Xinwang Liu** [1]   **Yue Liu** [1]   **Yi Zhang** [1]

## Abstract

Current multi-view clustering (MVC) techniques generally focus only on the relationship between anchors and samples, while overlooking that between anchors. Moreover, due to the lack of data labels, the cluster order is inconsistent across views and accordingly anchors encounter misalignment, which will confuse the graph structure and disorganize cluster representation. Even worse, it typically brings variance during forming spectral embedding, degenerating the stability of clustering results. In response to these concerns, in the paper we propose a MVC approach named DTP-SF-BVF. Concretely, we explicitly exploit the geometric properties between anchors via self-expression learning skill, and utilize topology learning strategy to feed captured anchor-anchor features into anchor-sample graph so as to explore the manifold structure hidden within samples more adequately. To reduce the misalignment risk, we introduce a permutation mechanism for each view to jointly rearrange anchors according to respective view characteristics. Besides not involving selecting the baseline view, it also can coordinate with anchors in the unified framework and thereby facilitate the learning of anchors. Further, rather than forming spectrum and then performing embedding partitioning, based on the criterion that samples and clusters should be hard assignment, we manage to construct the cluster labels directly from original samples using the binary strategy, not only preserving the data diversity but avoiding variance. Experiments on multiple publicly available datasets confirm the effectiveness of proposed DTP-SF-BVF method.

[1]School of Computer, National University of Defense Technololgy, Hunan, China. [2]Intelligent Game and Decision Lab, Beijing, China. yu-shengju@foxmail.com. Correspondence to: Siwei Wang <wangsiwei13@nudt.edu.cn>, Xinwang Liu <xinwangliu@nudt.edu.cn>.

*Proceedings of the 42$^{nd}$ International Conference on Machine Learning*, Vancouver, Canada. PMLR 267, 2025. Copyright 2025 by the author(s).

## 1. Introduction

In recent years, multi-view clustering (MVC) is becoming a research hotspot because of its ability to effectively mine potential patterns hidden in heterogeneous data, and is widespreadly deployed in various fields such as drug design and finance analysis (Xu et al., 2024; Yang et al., 2023a; Wang et al., 2023; Yu et al., 2024b; Wen et al., 2024a; Liang et al., 2023; Li et al., 2025; Ma et al., 2024b). As a powerful tool in MVC, anchor technique is commonly utilized to filter noise points and decrease the computing overhead (Li et al., 2023; Yu et al., 2025b; Zhang et al., 2025). It first selects a small number of significant samples to represent overall samples, and then replaces the sample-sample proximity relationship by building up the anchor-sample relationship (Ma et al., 2024a; Yang et al., 2022; Yu et al., 2024c). Following this line, a series of prominent works have been successively proposed. For instance, Kang et al. (2020b) regard the centroids generated by $k$-means on respective view as anchors and merge multiple graphs by splicing their left singular vectors. Xia et al. (2023) first project original samples to perform de-correlation and then select anchors in projection space according to the sample variance. Wang et al. (2022a) design a hierarchical $k$-means model to output anchors and construct sparse similarity using the learned bipartite graph. Unlike them, Huang et al. (2023) leverage three diversity levels in neighbors to construct anchors and generate graph directly in the early-stage fusion.

Although generating pleasing clustering results from various aspects, current methods usually focus only on the anchor-sample proximity relationship, and fail to take into account the anchor-anchor characteristics. This is not reasonable since between anchors, there generally exist informative geometric features. Overlooking them will not be conductive to constructing discriminative anchors and extracting the intrinsic similarity among samples. Additionally, due to the fact that clustering tasks do not involve any data labels, anchors could be misaligned across views, leading to the graph structure becoming chaotic. Wang et al. (2022b) provide an alignment scheme from the perspectives of feature and structure matching, nevertheless, it requires to select the baseline view. Also, the anchor generation, the anchor transformation, and the graph construction are separated from

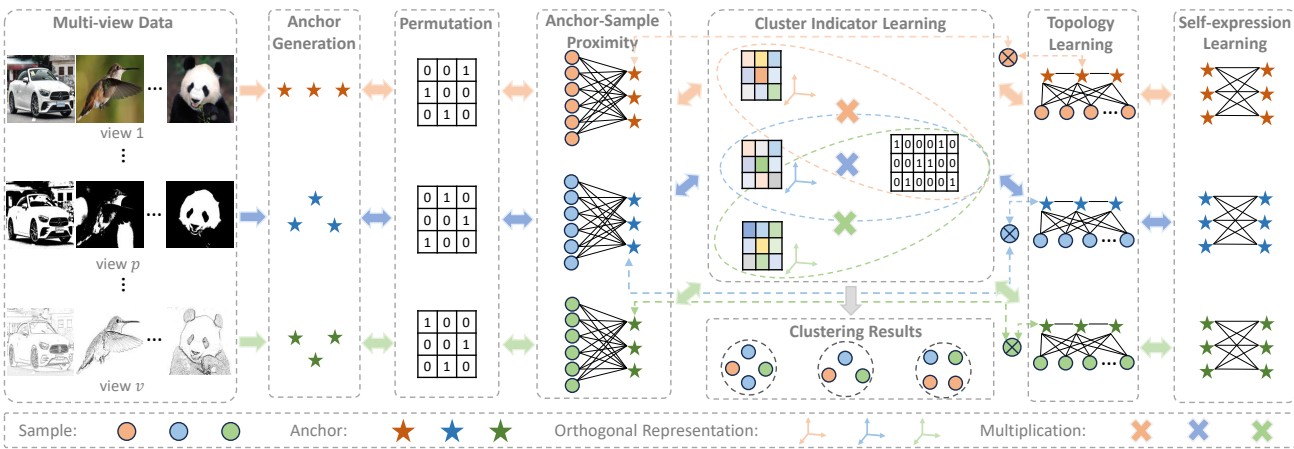

*Figure 1.* The devised DTP-SF-BVF multi-view clustering framework. It explicitly extracts the geometric characteristics of anchor-anchor via self-expression learning, and delivers them into the topology learning of anchor-sample so as to exploit the manifold structure among samples more sufficiently. It introduces a learnable permutation model for each view to alleviate the anchor misalignment. Instead of constructing spectrum and then conducting partitioning, it directly learns the cluster indicators via binary learning to avoid introducing variance. These three sub-parts are all jointly optimized within an unified learning model so as to move towards mutual reinforcement.

each other. These limitations hinder the interaction of view information across different levels and accordingly weaken the distinctiveness of anchors. Furthermore, the clustering procedure adopted by current methods is to first form spectrum and then conduct embedding partitioning on it, which causes the generated clustering results containing non-zero variance, degrading the stability and interpretability.

With these concerns in mind, we design a MVC method termed DTP-SF-BVF in this paper, and its framework is presented in Fig. 1. To be specific, we introduce self-expression learning mechanism to explore the geometric characteristics between anchors, and integrate them into the topology learning of anchor-sample graph so as to characterize the manifold structure inside samples more sufficiently. Then, we associate each view with a permutation model, which is learnable and works jointly with the anchor generation, to rearrange anchors in their original dimension space according to view-specific features. Owing to the joint-optimization mechanism in the unified framework, consequently, it does not involve the selection of baseline view. Further, to eliminate variance, based on the criterion that one sample should belong to only one cluster, we avoid the formation of spectrum and choose to directly generate cluster indicators from original samples. When the sample belongs to its cluster, we manage to optimize its indicator as 1 and otherwise 0. In addition to well preserving the data diversity, this paradigm also can skip the spectral partitioning stage and thereby alleviate the computing burden. The cluster indicator matrix is shared for all views, which bridges all anchors, permutations and views. Not only does it play an important role in gathering multi-view information at the cluster-label level,

but provides consensus structure for anchors on different views to force them rearranging towards correct-aligning direction. Subsequently, we give a six-step updating scheme with linear complexity to optimize the resultant objective loss. Experiments on multiple benchmark datasets demonstrate that DTP-SF-BVF is effective in grouping multi-view data and owns competitive strengths against multiple classical MVC approaches. For more clarity, we summary the contributions of this work as below,

1. We explicitly take into account the geometric features between anchors, and successfully integrate them into the anchor-sample graph through topology learning to exploit the manifold characteristics hidden within samples more fully for better clustering.

2. We devise a joint-alignment mechanism that not only eliminates the need for selecting the baseline view but also coordinates well with the generation of anchors.

3. We avoid the formation of spectrum by directly learning cluster indicators using a binary strategy, which effectively clears the variance in clustering results, accordingly highlighting the stability.

4. We provide a six-step optimization scheme with linear complexity for the loss function. Experiments validate the effectiveness of our proposed method from multiple aspects.

## 2. Related Work

Based on the fact that each view data typically owns self-unique features and consequently can compensate for the limitations of other views, multi-view clustering aims at

integrating information from diverse views to obtain more comprehensive and accurate data representation, thereby achieving superior clustering effect than single-view counterparts (Liang et al., 2024a; Yu et al., 2023a; Wang et al., 2024; Wan et al., 2024; Wen et al., 2023; Zhang et al.; Tang & Liu, 2022a; Yu et al., 2024e; Yang et al., 2024). Anchor technology is recently introduced into multi-view clustering to increase the computing efficiency (Shi et al., 2021; Yu et al., 2024d). It is intended to replace the full graph with a small-sized anchor graph by utilizing some discriminative landmarks. Specially, given a dataset $\{\mathbf{X}_p \in \mathbb{R}^{d_p \times n}\}_{p=1}^v$ where $d_p$, $n$ and $v$ denote the dimension of data, the number of samples and the number of views respectively, anchor based multi-view clustering can be formulated as

$$\min_{\Psi} \sum_{p=1}^v \|\mathbf{X}_p - \mathbf{A}_p\mathbf{Z}_p\|_F^2 + \eta \|\mathbf{Z}_p\|_F^2 + \gamma \|\mathbf{Z}_p - \mathbf{Z}\|_F^2 , \quad (1)$$

where $\Psi = \{\mathbf{Z}_p^\top\mathbf{1} = \mathbf{1}, \mathbf{Z}_p \geq 0, \mathbf{Z}^\top\mathbf{1} = \mathbf{1}, \mathbf{Z} \geq 0, \mathbf{A}_p \in \mathbb{R}^{d_p \times m}, \mathbf{Z}_p \in \mathbb{R}^{m \times n}\}_{p=1}^v$, $\eta$ and $\gamma$ denote the anchor matrix, anchor graph and regularization hyper-parameters, respectively. The fusion graph $\mathbf{Z} \in \mathbb{R}^{m \times n}$ aims at gathering the information from different views at the graph level. The non-negative constraints and column sum constraints guarantee the learned graph to satisfy the similarity requirements. After obtaining $\mathbf{Z}$, the cluster labels can be received by first constructing spectrum on the fusion graph $\mathbf{Z}$ and then conducting spectral partitioning operation on the embedding.

Noticed that the final clustering results are heavily dependent on the quality of $\mathbf{Z}_p$ while $\mathbf{Z}_p$ is related to anchor matrix $\mathbf{A}_p$, consequently, many works focus on the generation way of anchors. For example, Chen et al. (2023b) utilize tensor learning to investigate the low-rankness within views and employ a dynamic anchor learning strategy to explore that between views. Yan et al. (2022) integrate anchor learning and feature learning together, and learn to generate anchors separately. Given the fact that similar samples typically lie in the same cluster and have homologous characteristics, Li et al. (2022a) devise an alternative sampling scheme, which is independent of initialization, to generate anchors. Liu et al. (2024) narrow the distributions of anchors by leveraging the correlation information between views to enhance their distinction. These methods successfully construct representative anchors from different perspectives, nevertheless, they generally pay only attention to the relationship between anchors and samples when constructing anchor graph, while overlooking the influence of geometric characteristics inside anchors. This could bring about the loss of some informative features. Anchors on different views also could be misaligned due to the unsupervised property of data, leading to the confusion of graph structure (Wang et al., 2022b). Besides, the clustering results outputted by current approaches usually contain variance when partitioning the spectrum, which exacerbates the instability (Zhang

et al., 2020a; Zeng et al., 2024; Chen et al., 2023a). In next section, we will elaborate in detail on the principles of our devised DTP-SF-BVF approach to alleviate these issues.

## 3. Methodology

To explore the geometric properties between anchors, inspired by subspace reconstruction (Zhang et al., 2020b; Xia et al., 2022), we introduce self-expression learning for anchors. To be specific, we utilize the paradigm $\|\mathbf{A}_p - \mathbf{A}_p\mathbf{S}_p\|_F^2$ to explicitly extract the global structure between anchors. Especially, due to $\mathbf{S}_p \in \mathbf{R}^{m \times m}$ where $m$ is the number of anchors, solving $\mathbf{S}_p$ will take $\mathcal{O}(m^3)$ computing overhead, which is almost ignorable against $\mathcal{O}(m^2n)$ that solving $\mathbf{Z}_p$ takes since $m$ is far less than $n$. Then, to integrate the characteristics of anchor-anchor into anchor-sample so as to exploit the manifold features inside samples, we adopt the idea of point-point guidance to adjust the anchor graph. Specially, we utilize the element $[\mathbf{S}_p]_{i,j}$ to guide $[\mathbf{Z}_p]_{i,t}$ and $[\mathbf{Z}_p]_{j,t}$, $i,j = 1, \cdots, m$, $t = 1, \cdots n$, which can be formulated as $\sum_{i,j=1}^m \|[\mathbf{Z}_p]_{i,:} - [\mathbf{Z}_p]_{j,:}\|_2^2 [\mathbf{S}_p]_{i,j}$ and aims at restricting similar features to maintain the consistency. At this point, MVC objective can be devised as

$$\min_{\{\mathbf{Z}_p,\mathbf{S}_p\}_{p=1}^v} \sum_{p=1}^v \|\mathbf{X}_p - \mathbf{A}_p\mathbf{Z}_p\|_F^2 + \lambda \|\mathbf{A}_p - \mathbf{A}_p\mathbf{S}_p\|_F^2$$
$$+ \beta \sum_{i,j=1}^m \|[\mathbf{Z}_p]_{i,:} - [\mathbf{Z}_p]_{j,:}\|_2^2 [\mathbf{S}_p]_{i,j} \quad (2)$$

Subsequently, to eliminate the anchor misalignment issue, one straightforward idea is to compute the space similarity between anchor sets and then match anchors according to their distance. However, multi-view data generally has various dimensions, and accordingly anchors on different views also have various dimensions. It is typically difficult to directly compute the distance between anchor sets with diverse dimensions. Although one can project all anchors into a common space to make them have the same dimension, it can not guarantee the distance similarity after projecting to be consistent with that before projecting. Additionally, determining the appropriate projection dimension needs heuristic searching. The projecting operation also could lead to heavy information loss. Consequently, these strategies are not that sensible. To get rid of this dilemma, considering that the nature of anchor misalignment is that the order of anchors on different views is not identical, we can alleviate the misalignment issue by rearranging anchors. In particular, we associate each view with a learnable permutation matrix $\mathbf{T}_p \in \mathbb{R}^{m \times m}$ to flexibly transform anchors according to the characteristics of respective view, i.e., $\|\mathbf{X}_p - \mathbf{A}_p\mathbf{T}_p\mathbf{Z}_p\|_F^2$. The subsequent issue is how to make anchors rearrange towards the correct corresponding direction. Next, we solve this and the variance issue concurrently.

Due to variance arising from the construction of spectrum, we avoid forming spectrum, and choose to directly learn the cluster indicators. Especially, we factorize the anchor graph as a basic coefficient matrix and a consensus matrix, and utilize binary learning to optimize the consensus matrix. This not only makes the consensus matrix successfully represent the cluster indicators, but also provides a common structure for anchors on all views, inducing them rearranging towards the common structure. Further, since views typically own different levels of importance, we introduce a weighting variable for each view to automatically measure its contributions. Therefore, our DTP-SF-BVF is devised as

$$\min_{\boldsymbol{\Omega}} \sum_{p=1}^{v} \left\{ \boldsymbol{\alpha}_p^2 \|\mathbf{X}_p - \mathbf{A}_p \mathbf{T}_p \mathbf{B}_p \mathbf{C}\|_F^2 + \right.$$

$$\left. \lambda \|\mathbf{A}_p \mathbf{T}_p - \mathbf{A}_p \mathbf{T}_p \mathbf{S}_p\|_F^2 + \beta \operatorname{Tr}(\mathbf{B}_p^\top \mathbf{L_s} \mathbf{B}_p \mathbf{C} \mathbf{C}^\top) \right\}$$

$$\text{s.t. } \boldsymbol{\alpha}^\top \mathbf{1} = 1, \boldsymbol{\alpha} \geq 0, \mathbf{B}_p^\top \mathbf{B}_p = \mathbf{I}_k, \mathbf{T}_p^\top \mathbf{1} = \mathbf{1}, \mathbf{T}_p \mathbf{1} = \mathbf{1},$$

$$\mathbf{T}_p \in \{0,1\}^{m \times m}, \sum_{i=1}^{k} \mathbf{C}_{i,j} = 1, j = 1, 2, \ldots, n,$$

$$\mathbf{C} \in \{0,1\}^{k \times n}, \mathbf{S}_p^\top \mathbf{1} = \mathbf{1}, \mathbf{S}_p \geq 0, \sum_{i=1}^{m} [\mathbf{S}_p]_{i,i} = 0,$$

(3)

where $\boldsymbol{\Omega} = \{\mathbf{A}_p \in \mathbb{R}^{d_p \times m}, \mathbf{B}_p \in \mathbb{R}^{m \times k}, \mathbf{S}_p \in \mathbb{R}^{m \times m}, \mathbf{T}_p \in \mathbb{R}^{m \times m}, \boldsymbol{\alpha} \in \mathbb{R}^{v \times 1}, \mathbf{C} \in \mathbb{R}^{k \times n}; p = 1, \cdots, v\}$. The second term aims at capturing the characteristics between anchors. The third term is the matrix form of point-point guidance, and aims at delivering the characteristics of anchor-anchor into anchor-sample of the first term, where $\mathbf{L_s} \in \mathbb{R}^{m \times m} = \mathbf{D}_p - \mathbf{S}_p$, $\mathbf{D}_p = diag\{\sum_{j=1}^{m} [\mathbf{S}_p]_{i,j} \mid , i = 1, \cdots, m\}$. This spectrum-free model directly outputs discrete clustering results via the consensus cluster indicator matrix $\mathbf{C}$. $\boldsymbol{\alpha}$ plays a role in adjusting the importance between views. More illustrations for the objective function please refer to **Section** E in Appendix.

## 4. Solver

We adopt the alternating optimization idea to minimize the loss function Eq. (3).

**Update $\mathbf{A}_p$:** The optimization w.r.t $\mathbf{A}_p$ in Eq. (3) can be written as

$$\min_{\mathbf{A}_p} \boldsymbol{\alpha}_p^2 \|\mathbf{X}_p - \mathbf{A}_p \mathbf{T}_p \mathbf{B}_p \mathbf{C}\|_F^2 + \lambda \|\mathbf{A}_p \mathbf{T}_p - \mathbf{A}_p \mathbf{T}_p \mathbf{S}_p\|_F^2$$

(4)

By utilizing the derivative equal to zero, we can obtain

$$\mathbf{A}_p = \boldsymbol{\alpha}_v^2 \mathbf{X}_p \mathbf{E}_p^\top \left( \boldsymbol{\alpha}_v^2 \mathbf{E}_p \mathbf{E}_p^\top + \lambda \mathbf{F}_p \mathbf{F}_p^\top \right)^{-1}, \quad (5)$$

where $\mathbf{E}_p \in \mathbb{R}^{m \times n} = \mathbf{T}_p \mathbf{B}_p \mathbf{C}$, $\mathbf{F}_p \in \mathbb{R}^{m \times m} = \mathbf{T}_p - \mathbf{T}_p \mathbf{S}_p$.

**Update $\mathbf{T}_p$:** The optimization w.r.t $\mathbf{T}_p$ in Eq. (3) can be written as

$$\min_{\mathbf{T}_p} \boldsymbol{\alpha}_p^2 \|\mathbf{X}_p - \mathbf{A}_p \mathbf{T}_p \mathbf{B}_p \mathbf{C}\|_F^2 + \lambda \|\mathbf{A}_p \mathbf{T}_p - \mathbf{A}_p \mathbf{T}_p \mathbf{S}_p\|_F^2$$

$$\text{s.t. } \mathbf{T}_p^\top \mathbf{1} = \mathbf{1}, \mathbf{T}_p \mathbf{1} = \mathbf{1}, \mathbf{T}_p \in \{0,1\}^{m \times m}.$$

(6)

Expanding the objective by trace operation, Eq. (6) can be further equivalently transformed as

$$\min_{\mathbf{T}_p} \operatorname{Tr} \left( \mathbf{T}_p^\top \mathbf{G}_p \mathbf{T}_p \left( \lambda \mathbf{H}_p + \boldsymbol{\alpha}_p^2 \mathbf{M}_p - 2\lambda \mathbf{S}_p^\top \right) - 2\boldsymbol{\alpha}_p^2 \mathbf{T}_p^\top \mathbf{J}_p \right)$$

$$\text{s.t. } \mathbf{T}_p^\top \mathbf{1} = \mathbf{1}, \mathbf{T}_p \mathbf{1} = \mathbf{1}, \mathbf{T}_p \in \{0,1\}^{m \times m},$$

(7)

where $\mathbf{G}_p \in \mathbb{R}^{m \times m} = \mathbf{A}_p^\top \mathbf{A}_p$, $\mathbf{H}_p \in \mathbb{R}^{m \times m} = \mathbf{S}_p \mathbf{S}_p^\top$, $\mathbf{M}_p \in \mathbb{R}^{m \times m} = \mathbf{B}_p \mathbf{C} \mathbf{C}^\top \mathbf{B}_p^\top$ and $\mathbf{J}_p \in \mathbb{R}^{m \times m} = \mathbf{A}_p^\top \mathbf{X}_p \mathbf{C}^\top \mathbf{B}_p^\top$. Given the characteristics of feasible region, we can obtain the optimal $\mathbf{T}_p$ via traversal searching on the one-hot vectors $\{\mathbf{e}_i\}_{i=1}^m$.

**Update $\mathbf{B}_p$:** The optimization w.r.t $\mathbf{B}_p$ in Eq. (3) can be written as

$$\min_{\mathbf{B}_p} \operatorname{Tr} \left( \mathbf{B}_p^\top \left( \beta \mathbf{L_s} + \boldsymbol{\alpha}_p^2 \mathbf{Q}_p \right) \mathbf{B}_p \mathbf{C} \mathbf{C}^\top - 2\boldsymbol{\alpha}_p^2 \mathbf{C} \mathbf{X}_p^\top \mathbf{A}_p \mathbf{T}_p \mathbf{B}_p \right)$$

$$\text{s.t. } \mathbf{B}_p^\top \mathbf{B}_p = \mathbf{I}_k,$$

(8)

where $\mathbf{Q}_p \in \mathbb{R}^{m \times m} = \mathbf{T}_p^\top \mathbf{A}_p^\top \mathbf{A}_p \mathbf{T}_p$. Then, we split the feasible region into $[\mathbf{B}_p]_{:,i}^\top [\mathbf{B}_p]_{:,i} = 1$ and $[\mathbf{B}_p]_{:,i}^\top [\mathbf{B}_p]_{:,j} = 0, i \neq j$. Further, combined with the fact that $\mathbf{C} \mathbf{C}^\top$ is a diagonal matrix, Eq. (8) can be equivalently transformed as

$$\min_{[\mathbf{B}_p]_{:,j}} [\mathbf{B}_p]_{:,j}^\top \sum_{i=1}^{n} \mathbf{C}_{j,i} \left( \beta \mathbf{L_s} + \boldsymbol{\alpha}_p^2 \mathbf{Q}_p \right) [\mathbf{B}_p]_{:,j} +$$

$$\left[ -2\boldsymbol{\alpha}_p^2 \mathbf{C} \mathbf{X}_p^\top \mathbf{A}_p \mathbf{T}_p \right]_{j,:} [\mathbf{B}_p]_{:,j}$$

$$\text{s.t. } [\mathbf{B}_p]_{:,j}^\top \mathbf{I}_{m \times m} [\mathbf{B}_p]_{:,j} - 1 = 0,$$

$$[[\mathbf{B}_p]_{:,1}, \cdots, [\mathbf{B}_p]_{:,l \neq j}, \cdots, [\mathbf{B}_p]_{:,k}]^\top [\mathbf{B}_p]_{:,j} = \mathbf{0}_{(k-1) \times 1}.$$

(9)

It is a quadratically constrained quadratic programming and can be solved by current software.

**Update $\mathbf{S}_p$:** The optimization w.r.t $\mathbf{S}_p$ in Eq. (3) can be written as

$$\min_{\mathbf{S}_p} \operatorname{Tr} \left( \mathbf{S}_p^\top \mathbf{Q}_p \mathbf{S}_p + 2 \left( -\mathbf{Q}_p - \frac{\beta}{2\lambda} \mathbf{M}_p \right) \mathbf{S}_p \right)$$

$$\text{s.t. } \mathbf{S}_p^\top \mathbf{1} = \mathbf{1}, \mathbf{S}_p \geq 0, \sum_{i=1}^{m} [\mathbf{S}_p]_{i,i} = 0.$$

(10)

Noticed that the constraints can be equivalently transformed as $\boldsymbol{\Psi} = \{[\mathbf{S}_p]_{:,j}^\top \mathbf{1} = 1, 0 \leq [\mathbf{S}_p]_{:,j}, \mathbf{e}_j^\top [\mathbf{S}_p]_{:,j} = 0, j = 1, 2, \cdots, m\}$, and therefore Eq. (10) is further converted as

$$\min_{\boldsymbol{\Psi}} [\mathbf{S}_p]_{:,j}^\top \mathbf{Q}_p [\mathbf{S}_p]_{:,j} + 2 \left( -\mathbf{Q}_p - \frac{\beta}{2\lambda} \mathbf{M}_p \right)_{j,:} [\mathbf{S}_p]_{:,j}.$$

(11)

**Algorithm 1** The proposed DTP-SF-BVF

---

**Input**: Multi-view data $\{\mathbf{X}_p\}_{p=1}^v$, hyper-parameters $\lambda$, $\beta$.
**Output**: Discrete cluster indicator matrix $\mathbf{C}$.
**Initialize**: $\{\mathbf{A}_p, \mathbf{T}_p, \mathbf{B}_p, \mathbf{S}_p\}_{p=1}^v$, $\mathbf{C}$, $\boldsymbol{\alpha}$.

1: **repeat**
2:      Update $\mathbf{A}_p$ via Eq. (5)
3:      Update $\mathbf{T}_p$ via Eq. (7)
4:      Update $\mathbf{B}_p$ via Eq. (9)
5:      Update $\mathbf{S}_p$ via Eq. (11)
6:      Update $\mathbf{C}$ via Eq. (14)
7:      Update $\boldsymbol{\alpha}_p$ via Eq. (16)
8: **until** convergent

---

It is a quadratic programming and can be easily solved.

**Update $\mathbf{C}$**: The optimization w.r.t $\mathbf{C}$ in Eq. (3) can be written as

$$\min_{\mathbf{C}} \operatorname{Tr}\left(\mathbf{C}^\top \mathbf{W} \mathbf{C} - \mathbf{Z} \mathbf{C}\right)$$
$$\text{s.t.} \sum_{i=1}^k \mathbf{C}_{i,j} = 1, j = 1, 2, \ldots, n, \mathbf{C} \in \{0,1\}^{k \times n}, \quad (12)$$

where $\mathbf{W} \in \mathbb{R}^{k \times k} = \sum_{p=1}^v \boldsymbol{\alpha}_p^2 \mathbf{B}_p^\top \mathbf{T}_p^\top \mathbf{A}_p^\top \mathbf{A}_p \mathbf{T}_p \mathbf{B}_p + \beta \mathbf{B}_p^\top \mathbf{L}_s \mathbf{B}_p$, $\mathbf{Z} \in \mathbb{R}^{n \times k} = 2 \sum_{p=1}^v \boldsymbol{\alpha}_p^2 \mathbf{X}_p^\top \mathbf{A}_p \mathbf{T}_p \mathbf{B}_p$. The constraints indicate that there is only one non-zero element in each column of $\mathbf{C}$, and thus we can solve $\mathbf{C}$ column by column. Eq. (12) can be further transformed as

$$\min_{\mathbf{C}_{:,j}} \mathbf{C}_{:,j}^\top \mathbf{W} \mathbf{C}_{:,j} - \mathbf{Z}_{j,:} \mathbf{C}_{:,j}$$
$$\text{s.t.} \sum_{i=1}^k \mathbf{C}_{i,j} = 1, \mathbf{C}_{:,j} \in \{0,1\}^{k \times 1}. \quad (13)$$

The item $\mathbf{C}_{:,j}^\top \mathbf{W} \mathbf{C}_{:,j}$ means that it takes a certain diagonal element of $\mathbf{W}$, and $\mathbf{Z}_{j,:} \mathbf{C}_{:,j}$ takes a certain element of $\mathbf{Z}_{j,:}$. Therefore, we can determine the corresponding index of minimum by $l^* = \arg\min_l \mathbf{W}_{l,l} - \mathbf{Z}_{j,l}$, $l = 1, 2, \cdots, k$. Then, the value of $\mathbf{C}_{:,j}$ can be obtained by

$$\mathbf{C}_{i,j} = \begin{cases} 1, & i = l^*, \\ 0, & i \neq l^*, i = 1, 2, \cdots, k. \end{cases} \quad (14)$$

**Update $\boldsymbol{\alpha}$**: The optimization w.r.t $\boldsymbol{\alpha}$ in Eq. (3) is

$$\min_{\boldsymbol{\alpha}} \sum_{p=1}^v \boldsymbol{\alpha}_p^2 \|\mathbf{X}_p - \mathbf{A}_p \mathbf{T}_p \mathbf{B}_p \mathbf{C}\|_F^2 \text{ s.t. } \boldsymbol{\alpha}^\top \mathbf{1} = 1, \boldsymbol{\alpha} \geq 0. \quad (15)$$

Since the item $\frac{1}{b_p} = \|\mathbf{X}_p - \mathbf{A}_p \mathbf{T}_p \mathbf{B}_p \mathbf{C}\|_F^2$ is a constant for $\boldsymbol{\alpha}$, the optimal $\boldsymbol{\alpha}$ can be determined via Cauchy inequality. Thus, we have

$$\boldsymbol{\alpha}_p = \frac{b_p}{\sum_{p=1}^v b_p}. \quad (16)$$

Algorithm 1 summarizes the pipeline of DTP-SF-BVF.

## 5. Complexity Analysis

**Space complexity**    The space complexity of DTP-SF-BVF is mainly from optimization variables $\mathbf{A}_p$, $\mathbf{T}_p$, $\mathbf{B}_p$, $\mathbf{S}_p$, $\mathbf{C}$ and $\boldsymbol{\alpha}$, $p = 1, 2, \cdots, v$. According to the fact that $\mathbf{A}_p \in \mathbb{R}^{d_p \times m}$, $\mathbf{T}_p \in \mathbb{R}^{m \times m}$, $\mathbf{B}_p \in \mathbb{R}^{m \times k}$, $\mathbf{S}_p \in \mathbb{R}^{m \times m}$, $\mathbf{C} \in \mathbb{R}^{k \times n}$ and $\boldsymbol{\alpha} \in \mathbb{R}^{v \times 1}$, we have that storing them will require $\mathcal{O}(d_p m)$, $\mathcal{O}(m^2)$, $\mathcal{O}(mk)$, $\mathcal{O}(m^2)$, $\mathcal{O}(nk)$ and $\mathcal{O}(1)$ memory overhead, respectively. Thus, storing all optimization variables will take $\mathcal{O}(dm + m^2 v + mkv + nk)$ memory overhead where $d$ represents the data dimension sum of all views and is independent of the sample number $n$. Further, since the number of anchors $m$ is generally greater than or equal to the number of clusters $k$, we have $m^2 v \geq mkv$. Besides, considering that $m$ is generally much smaller than $n$ and is also independent of $n$, we have that the space complexity of proposed DTP-SF-BVF is $\mathcal{O}(nk)$, which is linearly related to the number of samples $n$.

**Time complexity**    The time complexity of DTP-SF-BVF is mainly from the updating of all optimization variables. When updating $\mathbf{A}_p$, constructing $\mathbf{E}_p$ and $\mathbf{F}_p$ will take $\mathcal{O}(m^2 k + mkn)$ and $\mathcal{O}(m^3)$ respectively. Constructing the item $\boldsymbol{\alpha}_v^2 \mathbf{E}_p \mathbf{E}_p^\top + \lambda \mathbf{F}_p \mathbf{F}_p^\top$ and solving its inverse will take $\mathcal{O}(m^2 n + m^3)$ and $\mathcal{O}(m^3)$ respectively. Thus, updating $\mathbf{A}_p$ will take $\mathcal{O}(m^2 k + mkn + m^2 n + m^3 + d_p nm + d_p m^2)$. When updating $\mathbf{T}_p$, constructing $\mathbf{G}_p$, $\mathbf{H}_p$, $\mathbf{M}_p$ and $\mathbf{J}_p$ will take $\mathcal{O}(d_p m^2)$, $\mathcal{O}(m^3)$, $\mathcal{O}(mkn + m^2 n)$ and $\mathcal{O}(d_p mn + mnk + m^2 k)$, respectively. Traversal searching on one-hot vectors will take $\mathcal{O}(m!)$. Thus, updating $\mathbf{T}_p$ will take $\mathcal{O}(d_p m^2 + d_p mn + m^3 + mkn + m^2 n + m!)$. When updating $\mathbf{B}_p$, constructing $\mathbf{Q}_p$ and the item $\mathbf{C} \mathbf{X}_p^\top \mathbf{A}_p \mathbf{T}_p \mathbf{B}_p$ will take $\mathcal{O}(d_p m^2)$ and $\mathcal{O}(knd_p + kd_p m + km^2 + k^2 m)$, respectively. Performing quadratically constrained quadratic programming will take $\mathcal{O}(m^3 k)$. Thus, updating $\mathbf{B}_p$ will take $\mathcal{O}(d_p m^2 + knd_p + k^2 m + m^3 k)$. When updating $\mathbf{S}_p$, due to the construction of $\mathbf{Q}_p$ and $\mathbf{M}_p$ having been completed, it only involves the performing of quadratic programming, which will take $\mathcal{O}(m^3)$. When updating $\mathbf{C}$, constructing $\mathbf{W}$ and $\mathbf{Z}$ will take $\mathcal{O}(d_p m^2 + d_p mk + d_p k^2 + km^2 + k^2 m)$ and $\mathcal{O}(nd_p m + nm^2 + nmk)$, respectively. Since the value of $\mathbf{C}$ can be determined by comparing the diagonal element of $\mathbf{W}$ and the row of $\mathbf{Z}$, updating $\mathbf{C}$ will take $\mathcal{O}(d_p mk + d_p k^2 + km^2 + k^2 m + nd_p m + nm^2 + nmk)$. When updating $\boldsymbol{\alpha}$, constructing $b_p$ will take $\mathcal{O}(d_p m^2 + d_p mk + d_p kn)$. The value of $\boldsymbol{\alpha}$ can be determined by Cauchy inequality, and thus updating $\boldsymbol{\alpha}$ will take $\mathcal{O}(d_p m^2 + d_p mk + d_p kn)$. Based on these, we have that updating all $\mathbf{A}_p$, $\mathbf{T}_p$, $\mathbf{B}_p$, $\mathbf{S}_p$, $\mathbf{C}$ and $\boldsymbol{\alpha}$ will take $\mathcal{O}(mknv + m^2 nv + dnm + dm^2 + m!v + m^3 kv + knd + k^2 mv + dk^2)$. Besides, considering that $m$ is usually greater than or equal to $k$, $d_p$ is independent of $n$, $n$ is largely greater than $m$, we can obtain that updating all variables will take $\mathcal{O}(m^2 nv + dnm + m!v + m^3 kv)$, which is also linearly related to the number of samples $n$.

# 6. Experiments

## 6.1. Experimental Setting

**Datasets** We evaluate the algorithm performance on the following 7 datasets: DERMATO, CALTE7, Cora, REU7200, Reuters, CIF10Tra4, FasMNI4V.

**Baselines** We choose the following 17 classical MVC methods as the baselines to demonstrate the effectiveness of proposed DTP-SF-BVF: FMR (Li et al., 2019), PMSC (Kang et al., 2020a), AMGL (Nie et al., 2016), MSCIAS (Wang et al., 2019), MVSC (Gao et al., 2015), MLRSSC (Brbić & Kopriva, 2018), MPAC (Kang et al., 2019), MCLES (Chen et al., 2020), FMCNOF (Yang et al., 2021), PFSC (Lv et al., 2021), SFMC (Li et al., 2022a), MSGL(Kang et al., 2022), FPMVS (Wang et al., 2022c), UOMVSC (Tang et al., 2023), PGSC(Wu et al., 2023), OrthNTF(Li et al., 2024b), FASTMI(Huang et al., 2023).

**Parameter Setup** We search the hyper-parameters $\lambda$ and $\beta$ in $[10^{-1}, 10^{0}, 10^{1}, 10^{2}, 10^{3}]$ and $[2^{-4}, 2^{-2}, 2^{0}, 2^{2}, 2^{4}]$ respectively. For all competitors, we download their source code and tune the parameters according to their provided guidelines. Three popular metrics are employed to measure the clustering results. For fairness, we run 20 times and calculate the mean and variance of clustering results.

## 6.2. Clustering Results and Analysis

The clustering results are reported in Table 1, and from this table we can conclude that,

1. ***Overall Effectiveness.*** Our DTP-SF-BVF consistently beats these competitors in terms of all three metrics on DERMATO, CALTE7, Reuters and FasMNI4V. Particularly, it makes 6.91% improvement in Fscore than the second-best approach on DERMATO. In other cases, such as on Cora, it is still able to provide comparable outcomes. These indicate that our DTP-SF-BVF is effective in partitioning multi-view data and can achieve competitive clustering outcomes.

2. ***Anchor Suitability.*** In contrast with PMSC, AMGL, MCLES, FMCNOF, OrthNTF, FMR, PGSC, etc, which tackle MVC problems using tensor, kernel, latent space, co-training or matrix factorization means, our DTP-SF-BVF using anchor tool can produce better results than them in most cases. For instance, on Cora, it surpasses them in terms of NMI with 38.59%, 42.37%, 27.00%, 42.63%, 43.20%, 23.07%, 42.55%, respectively. These suggest that our adopted anchor means is recommendable.

3. ***Ample Proximity.*** Different from FPMVS, FASTMI, SFMC, etc, which concentrate only on the anchor-sample proximity relationship, DTP-SF-BVF also successfully takes anchor-anchor characteristics into the measuring of overall similarity and accordingly brings performance enhancement. Taking FASTMI as an example, DTP-SF-BVF

*Table 1.* Clustering result comparison (mean±std)

| Dataset | DERMATO | CALTE7 | Cora | REU7200 | Reuters | CIF10Tra4 | FasMNI4V |
|---|---|---|---|---|---|---|---|
| ACC(%) | | | | | | | |
| FMR | 80.87(±5.92) | 39.95(±0.66) | 40.54(±1.98) | - | - | - | - |
| PMSC | 80.01(±9.96) | 49.92(±2.58) | 28.84(±0.74) | 23.57(±0.48) | - | - | - |
| AMGL | 22.75(±0.31) | 39.84(±2.06) | 14.99(±0.18) | 16.78(±0.01) | 17.43(±0.05) | - | - |
| MSCIAS | 83.60(±3.85) | 43.89(±2.15) | 51.64(±2.74) | 23.66(±0.42) | 34.23(±0.37) | - | - |
| MVSC | 55.69(±8.57) | 49.86(±2.26) | - | - | - | - | - |
| MLRSSC | 67.53(±5.04) | 57.26(±0.00) | 31.08(±0.00) | 18.62(±0.34) | - | - | - |
| MPAC | 81.84(±0.00) | 71.64(±0.00) | 40.21(±0.00) | 24.79(±0.00) | - | - | - |
| MCLES | 46.18(±2.15) | 40.47(±1.06) | 32.03(±2.33) | - | - | - | - |
| FMCNOF | 62.85(±5.32) | 71.98(±5.67) | 29.10(±2.74) | 22.92(±2.57) | - | 21.62(±1.83) | 41.51(±2.62) |
| PFSC | 52.27(±4.99) | 57.87(±5.43) | - | - | - | - | - |
| SFMC | 49.44(±0.00) | 67.71(±0.00) | 30.50(±0.00) | 15.86(±0.00) | 25.55(±0.00) | 9.98(±0.00) | - |
| MSGL | 73.46(±0.97) | - | - | 20.78(±0.28) | 42.65(±0.21) | 22.57(±0.43) | - |
| FPMVS | 78.33(±7.05) | 61.47(±1.35) | 37.12(±2.53) | 28.01(±1.20) | 51.82(±2.56) | 27.12(±0.79) | 52.86(±3.35) |
| UOMVSC | 77.65(±0.00) | 67.10(±0.00) | 44.72(±0.00) | 23.26(±0.00) | 36.28(±0.00) | - | - |
| PGSC | 70.08(±6.07) | 52.76(±3.07) | 29.19(±2.07) | 28.13(±1.88) | 42.47(±0.89) | - | - |
| OrthNTF | 82.43(±0.00) | 68.84(±0.00) | 47.76(±0.00) | 23.36(±0.00) | 47.96(±0.00) | 25.88(±0.00) | 53.27(±0.00) |
| FASTMI | 74.13(±3.36) | 53.34(±2.84) | 47.10(±4.07) | 22.95(±0.89) | 42.31(±3.17) | 25.58(±0.66) | 55.44(±2.25) |
| Ours | 85.47(±0.00) | 80.66(±0.00) | 52.44(±0.00) | 26.22(±0.00) | 54.26(±0.00) | 26.83(±0.00) | 57.36(±0.00) |
| NMI(%) | | | | | | | |
| FMR | 79.18(±3.85) | 44.81(±0.93) | 20.63(±1.21) | - | - | - | - |
| PMSC | 86.14(±4.84) | 44.93(±0.81) | 6.12(±0.67) | 4.02(±0.55) | - | - | - |
| AMGL | 4.56(±0.72) | 44.95(±2.07) | 2.74(±0.36) | 1.17(±0.00) | 1.02(±0.02) | - | - |
| MSCIAS | 80.74(±2.93) | 28.36(±1.86) | 42.16(±0.47) | 5.66(±0.35) | 12.98(±0.14) | - | - |
| MVSC | 53.68(±9.10) | 37.74(±2.22) | - | - | - | - | - |
| MLRSSC | 63.85(±4.83) | 12.11(±0.00) | 2.47(±0.00) | 2.89(±0.75) | - | - | - |
| MPAC | 80.50(±0.00) | 45.12(±0.00) | 23.56(±0.00) | 6.56(±0.00) | - | - | - |
| MCLES | 28.12(±1.27) | 27.33(±0.74) | 16.70(±2.10) | - | - | - | - |
| FMCNOF | 51.10(±4.83) | 41.78(±3.22) | 5.18(±0.02) | 3.21(±0.17) | - | 11.02(±1.13) | 44.82(±2.32) |
| PFSC | 55.85(±2.33) | 39.09(±2.55) | - | - | - | - | - |
| SFMC | 38.68(±0.00) | 45.10(±0.00) | 7.95(±0.00) | 12.82(±0.00) | 12.20(±0.00) | 2.90(±0.00) | - |
| MSGL | 64.40(±1.21) | - | - | 3.66(±0.03) | 20.73(±0.76) | 10.69(±0.23) | - |
| FPMVS | 81.78(±5.21) | 45.00(±1.10) | 13.56(±1.67) | 5.60(±0.66) | 30.23(±3.30) | 15.13(±1.16) | 58.10(±3.02) |
| UOMVSC | 88.24(±0.00) | 45.07(±0.00) | 21.26(±0.00) | 11.17(±0.00) | 19.03(±0.00) | - | - |
| PGSC | 66.61(±2.84) | 33.95(±3.07) | 15.92(±1.08) | 4.94(±0.80) | 22.16(±0.42) | - | - |
| OrthNTF | 52.33(±0.00) | 42.12(±0.00) | 39.74(±0.00) | 7.43(±0.00) | 28.87(±0.00) | 11.58(±0.00) | 58.83(±0.00) |
| FASTMI | 81.83(±6.01) | 45.05(±1.45) | 31.21(±2.89) | 7.67(±0.56) | 29.29(±1.85) | 12.85(±0.32) | 59.03(±0.41) |
| Ours | 89.97(±0.00) | 45.25(±0.00) | 43.70(±0.00) | 6.25(±0.00) | 31.87(±0.00) | 15.64(±0.00) | 59.21(±0.00) |
| Fscore(%) | | | | | | | |
| FMR | 76.45(±5.48) | 45.29(±1.63) | 27.83(±1.13) | - | - | - | - |
| PMSC | 80.22(±8.10) | 51.13(±2.49) | 27.48(±0.67) | 26.37(±0.63) | - | - | - |
| AMGL | 18.27(±0.12) | 40.47(±1.57) | 24.78(±0.02) | 28.51(±0.00) | 28.61(±0.00) | - | - |
| MSCIAS | 80.90(±4.37) | 42.80(±1.11) | 41.84(±1.26) | 21.42(±0.36) | 33.94(±0.08) | - | - |
| MVSC | 54.52(±9.73) | 48.53(±1.96) | - | - | - | - | - |
| MLRSSC | 63.90(±4.48) | 49.62(±0.00) | 28.87(±0.00) | 27.69(±0.41) | - | - | - |
| MPAC | 81.01(±0.00) | 67.25(±0.00) | 29.25(±0.00) | 24.29(±0.00) | - | - | - |
| MCLES | 39.10(±1.37) | 36.16(±0.62) | 28.95(±0.82) | - | - | - | - |
| FMCNOF | 56.89(±4.24) | 67.43(±5.73) | 29.89(±4.82) | 21.29(±3.14) | - | 19.83(±2.77) | 36.74(±3.63) |
| PFSC | 55.46(±4.05) | 62.75(±7.08) | - | - | - | - | - |
| SFMC | 42.90(±0.00) | 65.50(±0.00) | 30.20(±0.00) | 27.69(±0.00) | 34.04(±0.00) | 18.13(±0.00) | - |
| MSGL | 70.39(±0.76) | - | - | 24.59(±0.53) | 37.57(±0.27) | 16.37(±0.86) | - |
| FPMVS | 80.35(±6.83) | 62.09(±1.21) | 25.36(±1.03) | 22.96(±1.16) | 42.53(±1.92) | 20.31(±0.56) | 48.43(±2.66) |
| UOMVSC | 79.17(±0.00) | 67.85(±0.00) | 33.12(±0.00) | 28.47(±0.00) | 35.23(±0.00) | - | - |
| PGSC | 69.15(±5.23) | 55.84(±4.70) | 29.29(±1.46) | 24.88(±0.32) | 38.57(±0.85) | - | - |
| OrthNTF | 78.43(±0.00) | 65.63(±0.00) | 37.52(±0.00) | 24.77(±0.00) | 39.68(±0.00) | 16.74(±0.00) | 47.67(±0.00) |
| FASTMI | 76.98(±5.19) | 56.39(±2.93) | 35.19(±2.36) | 25.94(±1.21) | 39.20(±1.81) | 14.35(±0.29) | 50.21(±1.25) |
| Ours | 87.92(±0.00) | 78.12(±0.00) | 41.12(±0.00) | 28.55(±0.00) | 44.84(±0.00) | 20.64(±0.00) | 51.37(±0.00) |

outperforms it on all of these seven datasets and three metrics, which reveals that our double-track proximity strategy can help extract representations more sufficiently.

4. ***Reliable Stability.*** The results outputed by our DTP-SF-BVF are all not with variance. This mainly benefits from avoiding the generation of spectrum. Not only does the spectrum-free property enhance the result stability, but allows the labels to be directly derived from original data, well maintaining the diversity. Despite zero-variance for methods like MPAC, SFMC and OrthNTF, the low-rank constraint could damage potential graph structure, accordingly weakening their performance.

5. ***Broader Applicability.*** Some methods like PMSC, PFSC, MFLVC, AMGL, UOMVSC, MCLES, PGSC, etc, can not work with large-sized CIF10Tra4 and FasMNI4V due to the intensive complexities or self-limitations, while our proposed DTP-SF-BVF operates normally with its lower complexities and meanwhile can produce superior clustering outcomes. So, DTP-SF-BVF enjoys broader applicability.

Due to the space limit, more conclusions are presented in the **Section** H of Appendix.

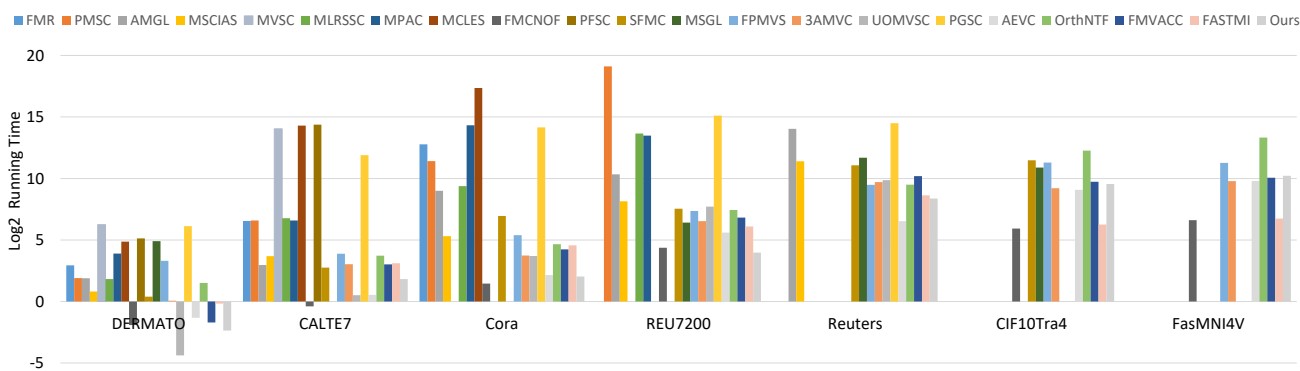

*Figure 2.* The running time comparison on seven public benchmark datasets.

To further exhibit the advantages of proposed DTP-SF-BVF, we also compare it with several popular deep learning competitors, ADAGAE (Li et al., 2022b), DEMVC (Xu et al., 2021a), MFLVC (Xu et al., 2022) and DSMVC (Tang & Liu, 2022b). Comparison results are summarized in Table 2.

*Table 2.* Comparison with deep learning based competitors

| Dataset | DERMATO | CALTE7 | Cora | REU7200 | Reuters | CIF10Tra4 | FasMNI4V |
|---|---|---|---|---|---|---|---|
| ACC(%) | | | | | | | |
| ADAGAE | 67.88(±0.99) | 42.20(±0.94) | 23.45(±0.29) | 19.43(±1.76) | - | - | - |
| DEMVC | 40.50(±0.88) | 54.41(±1.76) | 30.54(±1.64) | 24.58(±1.84) | 53.05(±0.91) | 26.75(±1.16) | 51.28(±0.95) |
| MFLVC | 80.73(±0.47) | 43.42(±0.26) | 31.02(±0.82) | 25.42(±1.47) | - | - | - |
| DSMVC | 72.35(±0.96) | 41.66(±1.30) | 28.88(±0.93) | 25.04(±0.47) | 53.66(±0.82) | 21.28(±0.99) | 55.33(±0.58) |
| Ours | 85.47(±0.00) | 80.66(±0.00) | 52.44(±0.00) | 26.22(±0.00) | 54.26(±0.00) | 26.83(±0.00) | 57.36(±0.00) |
| NMI(%) | | | | | | | |
| ADAGAE | 78.47(±0.36) | 39.28(±0.19) | 5.23(±0.68) | 3.22(±0.27) | - | - | - |
| DEMVC | 31.03(±0.11) | 16.70(±0.64) | 6.34(±0.30) | 4.84(±0.90) | 34.21(±0.35) | 16.18(±0.90) | 59.74(±0.91) |
| MFLVC | 81.23(±0.10) | 58.74(±0.15) | 12.97(±0.14) | 3.25(±0.90) | - | - | - |
| DSMVC | 76.15(±0.14) | 36.68(±0.18) | 8.14(±0.55) | 4.36(±0.41) | 35.43(±0.10) | 8.82(±0.46) | 55.33(±0.97) |
| Ours | 89.97(±0.00) | 45.25(±0.00) | 43.70(±0.00) | 6.25(±0.00) | 31.87(±0.00) | 15.64(±0.00) | 59.21(±0.00) |
| Fscore(%) | | | | | | | |
| ADAGAE | 67.74(±0.79) | 50.51(±0.41) | 23.68(±0.14) | 19.61(±1.23) | - | - | - |
| DEMVC | 41.80(±1.04) | 50.60(±1.59) | 27.66(±1.52) | 22.69(±1.14) | 56.39(±1.68) | 23.68(±1.57) | 48.39(±0.66) |
| MFLVC | 73.92(±1.63) | 52.68(±1.43) | 32.41(±1.05) | 25.13(±0.67) | - | - | - |
| DSMVC | 73.79(±1.89) | 51.00(±1.28) | 30.14(±1.12) | 25.01(±0.22) | 56.85(±1.54) | 21.01(±1.85) | 55.03(±1.86) |
| Ours | 87.92(±0.00) | 78.12(±0.00) | 41.12(±0.00) | 28.55(±0.00) | 44.84(±0.00) | 20.64(±0.00) | 51.37(±0.00) |

6. ***Competitive Ability.*** Against these competitors adopting deep learning technique to tackle MVC problems, we can produce preferable results in most cases, such as on DERMATO and Cora as well as REU7200. This illustrates that even comparing deep learning methods, our devised DTP-SF-BVF still provides competitive clustering results.

Then, we also conduct comparison with some alignment methods, FMVACC(Wang et al., 2022b), 3AMVC (Ma et al., 2024a) and AEVC (Liu et al., 2024) to illustrate the strength of our proposed baseline-view-free alignment strategy. Comparison results are presented in Table 3.

7. ***Flexible Alignment.*** Against FMVACC, 3AMVC and AEVC that require firstly selecting baseline view and then performs alignment, our results are more desirable in most cases. For example on CALTE7, we receive 27.32%, 6.84%, 37.11% improvement than FMVACC respectively. This is primarily because our joint-alignment property, besides not involving baseline view, also can coordinate with the generation of anchors, more flexibly transforming anchors.

*Table 3.* Comparison with alignment competitors

| Dataset | DERMATO | CALTE7 | Cora | REU7200 | Reuters | CIF10Tra4 | FasMNI4V |
|---|---|---|---|---|---|---|---|
| ACC(%) | | | | | | | |
| FMVACC | 74.13(±3.36) | 53.34(±2.84) | 47.10(±4.07) | 22.95(±0.89) | 42.31(±3.17) | 25.58(±0.66) | 55.44(±2.25) |
| 3AMVC | 62.60(±5.50) | 38.17(±3.58) | 44.73(±3.74) | 33.31(±1.19) | 49.56(±2.45) | 26.38(±0.93) | 57.24(±2.26) |
| AEVC | 91.82(±3.78) | 44.61(±4.31) | 39.68(±1.36) | 29.85(±0.02) | 50.88(±0.24) | 27.56(±1.11) | 54.89(±0.63) |
| Ours | 85.47(±0.00) | 80.66(±0.00) | 52.44(±0.00) | 26.22(±0.00) | 54.26(±0.00) | 26.83(±0.00) | 57.36(±0.00) |
| NMI(%) | | | | | | | |
| FMVACC | 80.78(±4.44) | 38.41(±2.92) | 33.50(±2.56) | 9.94(±1.54) | 28.50(±2.29) | 12.86(±0.67) | 57.82(±0.93) |
| 3AMVC | 57.43(±3.28) | 41.45(±4.42) | 26.63(±3.37) | 11.68(±2.11) | 31.03(±1.66) | 14.02(±0.71) | 58.61(±1.62) |
| AEVC | 86.62(±1.95) | 49.15(±0.86) | 17.79(±0.67) | 6.44(±0.02) | 24.47(±0.06) | 13.32(±0.50) | 53.55(±0.34) |
| Ours | 89.97(±0.00) | 45.25(±0.00) | 43.70(±0.00) | 6.25(±0.00) | 31.87(±0.00) | 15.64(±0.00) | 59.21(±0.00) |
| Fscore(%) | | | | | | | |
| FMVACC | 80.15(±7.13) | 41.01(±4.20) | 38.20(±1.89) | 23.79(±0.77) | 43.86(±2.61) | 17.07(±0.35) | 48.78(±1.94) |
| 3AMVC | 56.16(±4.80) | 38.28(±3.01) | 32.61(±2.53) | 27.58(±1.50) | 41.13(±1.30) | 17.40(±0.40) | 47.61(±1.62) |
| AEVC | 87.94(±3.70) | 46.16(±1.37) | 26.57(±1.24) | 22.19(±0.01) | 36.19(±0.63) | 17.15(±0.27) | 45.99(±0.53) |
| Ours | 87.92(±0.00) | 78.12(±0.00) | 41.12(±0.00) | 28.55(±0.00) | 44.84(±0.00) | 20.64(±0.00) | 51.37(±0.00) |

## 6.3. Running Time Comparison

To illustrate the efficiency of DTP-SF-BVF, we count the running time of each algorithm, and report the comparison results in Fig. 2. From this figure, we can draw that,

1. MVSC, PFSC, PGSC and MCLES consume significantly more time than others. This is mainly caused by the subspace strategy they employed, which typically requires constructing large-sized similarity and needs at least cubic computational overhead.

2. MPAC, PMSC, FMR, MLRSSC, etc, take more time than us, which is mainly because MPAC and PMSC gather multi-view representations at the partition level, and FMR and MLRSSC utilize the kernel dependence measure to do data reconstruction.

3. FPMVS, FMVACC, MSGL and SFMC operate slower than us. Possible reasons are that the connection component constraints and feature matching constraints conducted on anchor graph induce a large proportion of additional calculating expenditure.

4. FMCNOF and FASTMI enjoy slightly faster running speed, the reasons of which could be that FMCNOF decouples dense optimization matrices by sparse factorization skills and FASTMI generates base clusterings via fast parti-

tioning on the view-sharing graph.

5. AMGL, MSCIAS, UOMVSC and OrthNTF are generally faster than PMSC, MVSC, PGSC, MPAC, PFSC, etc, possibly because the former ones alleviate the computing burden of spectral partitioning and graph mergence via low-rank approximation or non-negative factorization.

6. All algorithms can normally work on DERMATO and CALTE7, while with the increase of sample size, PFSC, FMR, MCLES, PMSC, MPAC, MSGL, etc, are gradually ineffective, which is mainly due to the limitations of their innate computing requirement or memory cost.

### 6.4. Ablation Study

*Table 4.* The effectiveness of double-track proximity strategy

| Metric | Abla. | DERMATO | CALTE7 | Cora | REU7200 | Reuters | CIF10Tra4 | FasMNI4V |
|---|---|---|---|---|---|---|---|---|
| ACC | SLA | 71.51 | 49.05 | 30.35 | 16.75 | 47.05 | 26.69 | 52.15 |
| | DTP | **85.47** | **80.66** | **52.44** | **26.22** | **54.26** | **26.83** | **57.36** |
| NMI | SLA | 83.97 | 40.21 | 6.02 | 2.53 | 23.19 | 15.48 | 58.13 |
| | DTP | **89.97** | **45.25** | **43.70** | **6.25** | **31.87** | **15.64** | **59.21** |
| Fscore | SLA | 73.79 | 51.25 | 30.42 | 28.54 | 43.04 | 17.70 | 46.77 |
| | DTP | **87.92** | **78.12** | **41.12** | **28.55** | **44.84** | **20.64** | **51.37** |

To validate the effectiveness of double-track proximity (DTP), we organize relevant ablation (Abla.) experiments and present the comparison results in Table 4 where SLA denotes the clustering results of considering only anchor-sample relation. (We omit the variance items since they are all zero.) As seen, our DTP is coherently better than SLA, which well illustrates that the devised DTP strategy can help achieve superior results.

*Table 5.* The effectiveness of spectrum-free strategy

| Metric | Abla. | DERMATO | CALTE7 | Cora | REU7200 | Reuters | CIF10Tra4 | FasMNI4V |
|---|---|---|---|---|---|---|---|---|
| ACC | CS | 81.33($\pm$1.82) | 74.74($\pm$1.73) | 48.27($\pm$1.07) | 23.46($\pm$0.79) | **55.78($\pm$1.62)** | 22.64($\pm$1.07) | 55.03($\pm$0.92) |
| | SF | **85.47($\pm$0.00)** | **80.66($\pm$0.00)** | **52.44($\pm$0.00)** | **26.22($\pm$0.00)** | 54.26($\pm$0.00) | **26.83($\pm$0.00)** | **57.36($\pm$0.00)** |
| NMI | CS | 84.32($\pm$1.32) | 40.63($\pm$1.89) | 40.02($\pm$2.31) | 6.02($\pm$0.68) | 26.23($\pm$1.37) | 12.22($\pm$0.97) | **60.14($\pm$0.36)** |
| | SF | **89.97($\pm$0.00)** | **45.25($\pm$0.00)** | **43.70($\pm$0.00)** | **6.25($\pm$0.00)** | **31.87($\pm$0.00)** | **15.64($\pm$0.00)** | 59.21($\pm$0.00) |
| Fscore | CS | 79.67($\pm$2.07) | 71.37($\pm$1.13) | 35.92($\pm$1.96) | 22.82($\pm$1.02) | 41.03($\pm$0.93) | 19.26($\pm$1.63) | 46.86($\pm$0.87) |
| | SF | **87.92($\pm$0.00)** | **78.12($\pm$0.00)** | **41.12($\pm$0.00)** | **28.55($\pm$0.00)** | **44.84($\pm$0.00)** | **20.64($\pm$0.00)** | **51.37($\pm$0.00)** |
| Time(s) | CS | 0.83 | 9.81 | 11.53 | 51.24 | 892.17 | 2003.72 | 3850.35 |
| | SF | **0.20** | **3.53** | **4.07** | **15.73** | **330.21** | **746.76** | **1193.83** |

Table 5 summarizes the ablation results about the spectrum-free (SF) strategy, where CS denotes the clustering results containing spectrum as previous methods do. Evidently, in addition to owning the ability to generate preferable and stable clustering results, our SF also enjoys less time consuming. This gives evidence that the designed SF is more suitable for MVC problems.

In the paper we adopt a baseline-view-free (BVF) strategy to decrease the anchor misalignment risk. To demonstrate its effectiveness, we report the ablation results in Table 6, where UA denotes the clustering results without involving alignment. According to this table, it is easy to discover that BVF makes more favorable clustering results than UA, which well suggests that our BVF is functional.

*Table 6.* The effectiveness of baseline-view-free strategy

| Metric | Abla | DERMATO | CALTE7 | Cora | REU7200 | Reuters | CIF10Tra4 | FasMNI4V |
|---|---|---|---|---|---|---|---|---|
| ACC | UA | 80.73 | 76.59 | 31.65 | 16.67 | 45.29 | 25.91 | 53.68 |
| | BVF | **85.47** | **80.66** | **52.44** | **26.22** | **54.26** | **26.83** | **57.36** |
| NMI | UA | 82.53 | 39.55 | 35.41 | 3.32 | 24.77 | 15.30 | 56.47 |
| | BVF | **89.97** | **45.25** | **43.70** | **6.25** | **31.87** | **15.64** | **59.21** |
| Fscore | UA | 79.47 | 72.23 | 30.69 | 21.14 | 42.59 | 17.90 | 47.41 |
| | BVF | **87.92** | **78.12** | **41.12** | **28.55** | **44.84** | **20.64** | **51.37** |

To further show the merits of baseline-view-free strategy, we organize the experiments involving baseline view (BV). The comparison results are reported in Table 7. Evidently, our strategy indeed brings performance improvement. More in-depth analysis please refer to **Section J** in Appendix.

*Table 7.* Comparison between baseline-view-free strategy and BV

| Dataset | DERMATO | CALTE7 | Cora | REU7200 | Reuters | CIF10Tra4 | FasMNI4V |
|---|---|---|---|---|---|---|---|
| | ACC(%) | | | | | | |
| BV | 72.37 | 62.43 | 45.78 | 22.87 | 44.36 | 22.98 | 51.22 |
| Ours | **85.47** | **80.66** | **52.44** | **26.22** | **54.26** | **26.83** | **57.36** |
| | NMI(%) | | | | | | |
| BV | 71.97 | 41.24 | 34.76 | 5.78 | 27.31 | **15.73** | 52.73 |
| Ours | **89.97** | **45.25** | **43.70** | **6.25** | **31.87** | 15.64 | **59.21** |
| | Fscore(%) | | | | | | |
| BV | 70.38 | 59.32 | 33.47 | 25.46 | 39.84 | 17.96 | 46.31 |
| Ours | **87.92** | **78.12** | **41.12** | **28.55** | **44.84** | **20.64** | **51.37** |

More ablation studies are placed into **Section I** of Appendix.

## 7. Limitations

DTP-SF-BVF contains hyper-parameters $\lambda$ and $\beta$, which requires additional efforts for fine-tuning. Consequently, designing a non-parametric version can further boost its practicality. Besides, we adopt the square weighting scheme with linear constraints to measure the contributions between views. Some other ingenious view schemes could be deeply investigated in the future so as to further increase the results.

## 8. Conclusion

In this work, we introduce double-track proximity, which concurrently considers anchor-sample and anchor-anchor characteristics, to more fully extract multi-view representations for better clustering. To reduce the mismatching risk, we adopt a joint-alignment mechanism that does not involve the selection of baseline view and also can coordinate with the anchor generation. Furthermore, we avoid forming spectrum and directly generate cluster indicators via a binary learning strategy, which not only effectively eliminates the variance but well preserves original diversity. For the resulting optimization problem, we provide a solution with linear complexities. Experiments on multiple public benchmark datasets verify the effectiveness of DTP-SF-BVF. In future work, we will extend our DTP-SF-BVF method to non-parametric scenarios to further enhance its practicality.

## Impact Statement

This paper presents work whose goal is to advance the field of Machine Learning. There are many potential societal consequences of our work, none which we feel must be specifically highlighted here.

## Acknowledgments

This work was supported by the National Natural Science Foundation of China under Grant No. 62406329, 62476280, 62441618, 62325604, 62276271.

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

## A. Notations

For more clarity, we summary the utilized symbols and their corresponding meaning, as shown in Table 8.

*Table 8.* The description of symbols used in this article

| Symbol | Meaning |
|---|---|
| $n$ | the number of samples |
| $m$ | the number of anchors |
| $v$ | the number of views |
| $k$ | the number of clusters |
| $d_p$ | the data dimension on view $p$ |
| $\mathbf{X}_p \in \mathbb{R}^{d_p \times n}$ | the data matrix on view $p$ |
| $\mathbf{A}_p \in \mathbb{R}^{d_p \times m}$ | the anchor matrix on view $p$ |
| $\mathbf{T}_p \in \mathbb{R}^{m \times m}$ | the permutation matrix on view $p$ |
| $\mathbf{B}_p \in \mathbb{R}^{m \times k}$ | the basic coefficient matrix on view $p$ |
| $\mathbf{C} \in \mathbb{R}^{k \times n}$ | the cluster indicator matrix |
| $\mathbf{S}_p \in \mathbb{R}^{m \times m}$ | the anchor self-expression matrix on view $p$ |
| $\mathbf{D}_p \in \mathbb{R}^{m \times m}$ | the degree matrix of $\mathbf{S}_p$ on view $p$ |
| $\boldsymbol{\alpha} \in \mathbb{R}^{v \times 1}$ | the view weighting vector |
| $\mathbf{Z}_p \in \mathbb{R}^{m \times n}$ | the anchor graph on view $p$ |
| $\mathbf{L_s} \in \mathbb{R}^{m \times m}$ | the Laplacian matrix about $\mathbf{S}_p$ |
| $\mathbf{E}_p \in \mathbb{R}^{m \times n}$ | $\mathbf{T}_p\mathbf{B}_p\mathbf{C}$ |
| $\mathbf{F}_p \in \mathbb{R}^{m \times m}$ | $\mathbf{T}_p - \mathbf{T}_p\mathbf{S}_p$ |
| $\mathbf{G}_p \in \mathbb{R}^{m \times m}$ | $\mathbf{A}_p^\top \mathbf{A}_p$ |
| $\mathbf{H}_p \in \mathbb{R}^{m \times m}$ | $\mathbf{S}_p\mathbf{S}_p^\top$ |
| $\mathbf{M}_p \in \mathbb{R}^{m \times m}$ | $\mathbf{B}_p\mathbf{C}\mathbf{C}^\top\mathbf{B}_p^\top$ |
| $\mathbf{J}_p \in \mathbb{R}^{m \times m}$ | $\mathbf{A}_p^\top \mathbf{X}_p\mathbf{C}^\top\mathbf{B}_p^\top$ |
| $\mathbf{Q}_p \in \mathbb{R}^{m \times m}$ | $\mathbf{T}_p^\top\mathbf{A}_p^\top\mathbf{A}_p\mathbf{T}_p$ |
| $\mathbf{Z} \in \mathbb{R}^{n \times k}$ | $2\sum_{p=1}^v \alpha_p^2\mathbf{X}_p^\top\mathbf{A}_p\mathbf{T}_p\mathbf{B}_p$ |
| $\mathbf{W} \in \mathbb{R}^{k \times k}$ | $\sum_{p=1}^v \alpha_p^2\mathbf{B}_p^\top\mathbf{T}_p^\top\mathbf{A}_p^\top\mathbf{A}_p\mathbf{T}_p\mathbf{B}_p + \beta\mathbf{B}_p^\top\mathbf{L_s}\mathbf{B}_p$ |

## B. Brief Introduction of Seventeen Comparison Methods

To demonstrate the strong points of the proposed DTP-SF-BVF, we select seventeen remarkable MVC methods as the baselines. Their brief introduction is as follows,

1. **FMR (Li et al., 2019):** This method utilizes kernel dependence measure instead of projecting original samples to enhance the correlation between different views, and highlights the comprehensiveness of potential representations through subspace reconstruction.

2. **PMSC (Kang et al., 2020a):** This method merges view information in the level of partition spaces via ensemble learning, and integrates consensus clustering and graph generation to maintain the consistence among views.

3. **AMGL (Nie et al., 2016):** This method assigns a group of weights for all graphs to increase the diversity automatically, and reformulates conventional spectral partitioning procedure into a convex problem so as to generate the optimal solution.

4. **MSCIAS (Wang et al., 2019):** This method maximizes the dependence between intact points by constructing an informative affinity matrix, and avoids view information imbalance by guiding intactness-aware relationship construction using HSIC criterion.

5. **MVSC (Gao et al., 2015):** This method conducts subspace clustering action on each view concurrently to explore specific characteristics, and employs an indicator matrix that is shared for all the views to preserve the cluster consistence.

6. **MLRSSC (Brbić & Kopriva, 2018):** This method generates a common similarity matrix with low-rank and sparsity properties to learn joint subspace representations, and utilizes the kernel extension skill to optimize the objective in Hilbert space.

7. **MPAC (Kang et al., 2019):** This method aligns each partition alternatively using a permutation matrix to formulate agreement cluster indicator, and performs graph learning and data partitioning jointly in one common framework to facilitate each other.

8. **MCLES (Chen et al., 2020):** This method tries to capture global structure by exploring embedding representations in latent space, and concurrently learns the cluster labels and similarity matrix without requiring subsequent spectral grouping procedure.

9. **FMCNOF (Yang et al., 2021):** This method integrates matrix factorization skill and bipartite graph construction together to improve the computational overhead, and embeds the factor matrix into cluster matrix to avoid extra $k$-means operation.

10. **PFSC (Lv et al., 2021):** This method finds a common partition by collaboratively learning multiple basic partitions to improve the robustness to noise, and jointly performs basic partition generation and unified graph learning to achieve mutual co-evolution.

11. **SFMC (Li et al., 2022a):** This method coalesces view-specific costs to seek for a joint graph that is compatible among views, and indicates clusters straightforwardly by employing connectivity constraint on the joint graph.

12. **MSGL (Kang et al., 2022):** This method discriminates landmarks by building a dictionary matrix to decrease the cost of graph generation, and discovers a graph with explicit components to preserve the data manifold.

13. **FPMVS (Wang et al., 2022c):** This method designs a group of space-guided projection matrices to alleviate the dimension inconsistency in common space, and determines the contribution of each individual view to the unified graph in a learnable manner.

14. **UOMVSC (Tang et al., 2023):** This method unifies the spectral embedding and spectral discretization via one-pass strategy to alleviate the information loss caused by the two-step process, and approximates the rank of affinity graph through the inner product of embedding matrices.

15. **PGSC (Wu et al., 2023):** This method exploits the connectivity and sparsity of each similarity graph to achieve the pure graph with a block-diagonal structure, and assigns labels directly by enforcing it including corresponding connection components.

16. **OrthNTF (Li et al., 2024b):** This method establishes an orthogonal non-negative tensor factorization scheme to directly consider the cross-correlation between views, and extracts complementary information hidden in multi-view samples through tensor regularization.

17. **FASTMI (Huang et al., 2023):** This method achieves multi-stage mergence by building view-wise relations using random view grouping, and utilizes a graph partitioning mechanism to generate basic clusterings for each view group.

To further demonstrate the advantages of DTP-SF-BVF, we also compare it with several deep learning based competitors. Their introduction is as follows,

1. **ADAGAE (Li et al., 2022b):** This method utilizes a graph auto-encoder to extract the potential high-level information behind data and the non-euclidean structure, and avoids the collapse by building the connections between sub-clusters before they become thoroughly random in the latent space.

2. **DEMVC (Xu et al., 2021a):** This method generates the embedded feature representations by deep auto-encoders, and adopts the auxiliary distribution generated by $k$-means to refine the deep auto-encoders and clustering soft assignments for all views.

3. **MFLVC (Xu et al., 2022):** This method jointly realizes view-specific reconstruction objective and semantic consistency objectives by learning diverse levels of representations in a fusion-free way, and utilizes the common semantics to generate the clustering labels.

4. **DSMVC (Tang & Liu, 2022b):** This method concurrently exploits complementary information and discards the meaningless noise by automatically selecting features to reduce the risk of clustering performance degradation caused by view increase.

We also compare DTP-SF-BVF with popular alignment methods, and their introduction is as follows,

1. **FMVACC (Wang et al., 2022b):** This method utilizes feature information and structure information of the bipartite graph generated by fixed anchors to build the matching relationship, and regards the first view of each dataset as the baseline view.

2. **3AMVC (Ma et al., 2024a):** This method gets rid of prior knowledge by identifying and selecting discriminative anchors within a single view using hierarchical searching, and takes the view exhibiting the highest anchor graph quality as the baseline view.

3. **AEVC (Liu et al., 2024):** This method narrows the spatial distribution of anchors on similar views by leveraging the inter-view correlations to enhance the expression ability of anchors, and treats the view concatenated by column as the baseline view.

## C. Brief Introduction of Seven Public Datasets

In experiments, we evaluate the algorithm performance on seven public multi-view datasets, and their brief introduction is as follows,

1. **DERMATO:** This is a skin image dataset and consists of 358 samples. It contains 2 views and 6 clusters. The feature dimensions on each view are 12 and 22 respectively.

2. **CALTE7:** This is an object image dataset and consists of 1474 samples. It contains 6 views and 7 clusters. The feature dimensions on each view are 48, 40, 254, 1984, 512 and 928, respectively.

3. **Cora:** This citation network dataset has 2708 samples, and includes 4 views and 7 clusters. The feature dimensions on each view are 2708, 1433, 2708 and 2708, respectively.

4. **REU7200:** This document dataset has 7200 samples, and includes 5 views and 6 clusters. The feature dimensions on each view are 4819, 4810, 4892, 4858 and 4777, respectively.

5. **Reuters:** This is a news article dataset with 18758 samples, and involves 5 views and 6 clusters. The feature dimensions on each view are 21531, 24892, 34251, 15506 and 11547, respectively.

6. **CIF10Tra4:** This is a color image dataset with 50000 samples, and involves 4 views and 10 clusters. The feature dimensions on each view are 944, 576, 512 and 640, respectively.

7. **FasMNI4V:** This is a fashion product image dataset with 70000 samples, and involves 4 views and 10 clusters. The feature dimensions on each view are 512, 576, 640 and 944, respectively.

## D. Other Related Work

To effectively tackle MVC tasks, Chen et al. (2022) utilize the algebraic property to learn a group of orthogonal bases for anchors while preserving the scalability, Qiang et al. (2021) iteratively partition original data into two balanced parts using k-means++ to output informative anchors, Zhang et al. (2023) integrate anchor selection into the generation of anchor

graph in which the number of connection components is the same as that of clusters to explicitly explore cluster structure, Li et al. (2024c) devise a pre-defined prior matrix for view-wise anchors to regularize their order and utilize a graph matching model to handle unpaired data, Yu et al. (2023b) combine membership learning and the construction of anchors to decrease the disagreement between views, and improve the clearness of cluster grouping via trace norm regularizer, Lao et al. (2024) choose to jointly construct multiple sets of anchors for basic clusterings so as to form discriminative subspace representations.

Orthogonal to them, Xu et al. (2021b) optimize a view-common variable and view-specific variables by introducing variational auto encoder into MVC to regulate consecutive visual characteristics of multiple views, Cui et al. (2024) highlight consistent representations from the perspective of information theory and decrease the view redundancy by minimizing the representation lower bound, Zhang et al. (2022) reach to the balance between complementarity and consistency by encoding view information using an adversarial strategy and utilize a parameter-free loss to complete the formation of structured representations while avoiding over-fitting, Fu et al. (2024) excavate potential structure distributions among samples in a generative manner and utilize anchor graphs to guide the learning process by generating structured spectral embedding using graph convolution network. By virtue of tensor tool, Li et al. (2024a) orthogonally project anchor graph into a potential label space to explore the cluster distribution and alleviate the loss of spatial structure information caused by projection transformation via tensor regularization. Long et al. (2024) form an embedding tensor by stacking embedding features of all views together to simultaneously explore the inter-view and intra-view correlations, and utilize the uniformity between semantics by employing an unified constraint to guarantee the smoothness of embedding.

To enhance the block structure of anchor graph, Qin et al. (2022) integrate multiple similarity matrices into one by introducing semi-supervised information and concurrently perform self-mapping and backward encoding via reconstruction. Nie et al. (2024) conduct number limitations on each cluster by combining min-cut and size constraints to enhance the flexibility and decrease the parameter sensitivity, and decompose lower constraints and upper constraints respectively via augmented Lagrangian multiplier strategy. Wen et al. (2024b) enhance the robustness by reducing the negative impact of noisy features and redundant information using feature weighting constraints, and utilize graph-embedded learning to maintain the structure characteristics. Huang et al. (2022) construct various metrics by randomizing exponential similarity in metric subspace rather than original space to improve the diversification of similarity matrices, and probe into the spatial characteristics of clusters via entropy criteria. Zeng et al. (2023) capture unified semantics by eliminating the discrepancy across views using the semantically-invariant distribution hidden within views, and alleviate the impact of defective instances via distribution transformation skills. The approaches, such as (Lu et al., 2024; Yu et al.; Sun et al., 2024; Yu et al., 2024a; Xu et al., 2021a; Yang et al., 2023b; Yu et al., 2025a; Liang et al., 2024b), have been also studied.

## E. More Illustrations for the Objective Function

For the methodology, we here provide more details to explain the reasons for the design of each component.

First of all, to exploit the geometric characteristics between anchors, inspired by the concept of subspace reconstruction, we introduce self-expression learning for anchors. Specially, we utilize the paradigm $\|\mathbf{A}_p - \mathbf{A}_p \mathbf{S}_p\|_F^2$ to explicitly extract the global structure between anchors.

After obtaining the anchor-anchor characteristic $\mathbf{S}_p \in \mathbb{R}^{m \times m}$, we need to integrate that into anchor-sample so as to exploit the manifold features inside samples. To this end, we adopt the idea of point-point guidance to adjust the anchor graph. Note that the rows of anchor graph $\mathbf{Z}_p \in \mathbb{R}^{m \times n}$ correspond to anchors, and thus we utilize the element $[\mathbf{S}_p]_{i,j}$ to guide $[\mathbf{Z}_p]_{i,t}$ and $[\mathbf{Z}_p]_{j,t}$, $t = 1, \cdots n$, which can be formulated as $\sum_{i,j=1}^{m} \|[\mathbf{Z}_p]_{i,:} - [\mathbf{Z}_p]_{j,:}\|_2^2 [\mathbf{S}_p]_{i,j}$ and aims at restricting similar features to maintain the consistency.

Then, to alleviate the anchor misalignment, considering that the nature of misalignment is that the order of anchors on different views is not identical, we alleviate the misalignment issue by rearranging anchors. Particularly, we associate each view with a learnable permutation matrix $\mathbf{T}_p$ to freely transform anchors in the original dimension space according to the characteristics of respective view. In addition to not involving selecting the baseline view, our mechanism also can coordinate with anchors in the unified framework and thereby facilitates the learning of anchors. Correspondingly, the anchor matrix $\mathbf{A}_p$ is reformulated as $\mathbf{A}_p \mathbf{T}_p$. The self-expression term $\|\mathbf{A}_p - \mathbf{A}_p \mathbf{S}_p\|_F^2$ and the reconstruction term $\|\mathbf{X}_p - \mathbf{A}_p \mathbf{Z}_p\|_F^2$ are reformulated as $\|\mathbf{A}_p \mathbf{T}_p - \mathbf{A}_p \mathbf{T}_p \mathbf{S}_p\|_F^2$ and $\|\mathbf{X}_p - \mathbf{A}_p \mathbf{T}_p \mathbf{Z}_p\|_F^2$, respectively.

Subsequently, considering that the variance arises from the construction of spectrum, we avoid forming spectrum and choose to directly learn the cluster indicators. In particular, we factorize the anchor graph as a basic coeffi-

cient matrix $\mathbf{B}_p$ and a consensus matrix $\mathbf{C}$, and utilize binary learning to optimize the consensus matrix. Therefore, we have that the term $\|\mathbf{X}_p - \mathbf{A}_p \mathbf{T}_p \mathbf{Z}_p\|_F^2$ is reformulated as $\|\mathbf{X}_p - \mathbf{A}_p \mathbf{T}_p \mathbf{B}_p \mathbf{C}\|_F^2$. The point-point guidance term $\sum_{i,j=1}^m \|[\mathbf{Z}_p]_{i,:} - [\mathbf{Z}_p]_{j,:}\|_2^2 [\mathbf{S}_p]_{i,j}$ is reformulated as $\sum_{i,j=1}^m \|[\mathbf{B}_p\mathbf{C}]_{i,:} - [\mathbf{B}_p\mathbf{C}]_{j,:}\|_2^2 [\mathbf{S}_p]_{i,j}$, which can be equivalently transformed as the matrix trace form of $\mathrm{Tr}(\mathbf{B}_p^\top \mathbf{L_s} \mathbf{B}_p \mathbf{C}\mathbf{C}^\top)$. $\mathbf{L_s} = \mathbf{D}_p - \mathbf{S}_p$, $\mathbf{D}_p = diag\{\sum_{j=1}^m [\mathbf{S}_p]_{i,j} \mid, i = 1, \cdots, m\}$. This paradigm not only makes the consensus matrix $\mathbf{C}$ successfully represent the cluster indicators, but also provides a common structure for anchors on all views, inducing them rearranging towards the corresponding matching relationship.

At the last, due to views generally having different levels of importance, we assign a weighting variable to each view to adaptively adjust its contributions. Accordingly, the term $\|\mathbf{X}_p - \mathbf{A}_p\mathbf{T}_p\mathbf{B}_p\mathbf{C}\|_F^2$ is further reformulated as $\alpha_p^2 \|\mathbf{X}_p - \mathbf{A}_p\mathbf{T}_p\mathbf{B}_p\mathbf{C}\|_F^2$.

Based on the above analysis, we have that the objective is formulated as $\sum_{p=1}^v \alpha_p^2 \|\mathbf{X}_p - \mathbf{A}_p\mathbf{T}_p\mathbf{B}_p\mathbf{C}\|_F^2 + \lambda \|\mathbf{A}_p\mathbf{T}_p - \mathbf{A}_p\mathbf{T}_p\mathbf{S}_p\|_F^2 + \beta \mathrm{Tr}(\mathbf{B}_p^\top \mathbf{L_s}\mathbf{B}_p\mathbf{C}\mathbf{C}^\top)$.

The first item in Eq. (3) aims at building the similarity via minimizing the reconstruction error. The second item represents the self-expression affinity of aligned anchors. The third item plays a role in feeding anchor-anchor characteristics into anchor-sample.

Further, for the feasible region, the constraints $\{\boldsymbol{\alpha}^\top \mathbf{1} = 1, \boldsymbol{\alpha} \geq 0\}$ aim at doing normalization and meanwhile avoid trivial solutions. $\{\mathbf{B}_p^\top \mathbf{B}_p = \mathbf{I}_k\}$ aims at learning discriminative basic coefficients. $\{\mathbf{T}_p^\top \mathbf{1} = \mathbf{1}, \mathbf{T}_p\mathbf{1} = \mathbf{1}, \mathbf{T}_p \in \{0,1\}^{m \times m}\}$ aims at rearranging anchors and meanwhile guarantees not to change the values of anchors. $\{\sum_{i=1}^k \mathbf{C}_{i,j} = 1, j = 1, 2, \ldots, n, \mathbf{C} \in \{0,1\}^{k \times n}\}$ guarantees that there is only one non-zero element in each column, i.e., one sample belongs to only one cluster. $\{\mathbf{S}_p^\top \mathbf{1} = \mathbf{1}, \mathbf{S}_p \geq 0, \sum_{i=1}^m [\mathbf{S}_p]_{i,i} = 0\}$ guarantees expressing oneself through other anchors while avoiding using oneself to express oneself.

## F. The Influence of Anchor Number

In experiments, we set the number of anchors to be equal to the number of clusters. The reasons are as follows.

When updating the variable $\mathbf{T}_p$, the objective function is $\mathrm{Tr}\left(\mathbf{T}_p^\top \mathbf{G}_p\mathbf{T}_p\left(\lambda\mathbf{H}_p + \alpha_p^2\mathbf{M}_p - 2\lambda\mathbf{S}_p^\top\right) - 2\alpha_p^2\mathbf{T}_p^\top\mathbf{J}_p\right)$, which is the form of $\mathbf{A}^\top\mathbf{BAC} + \mathbf{A}^\top\mathbf{D}$. Besides, the feasible region $\{\mathbf{T}_p^\top\mathbf{1} = \mathbf{1}, \mathbf{T}_p\mathbf{1} = \mathbf{1}, \mathbf{T}_p \in \{0,1\}^{m \times m}\}$ is discrete. These cause this optimization problem being hard to solve. To this end, we adopt traversal searching on one-hot vectors to obtain the optimal solution. The traversal searching operation takes $\mathcal{O}(m!)$ computing overhead where $m$ is the number of anchors. Too large $m$ will induce intensive time cost. Therefore, in all experiments, we set $m$ to the number of anchors $k$. More genius solving schemes could be further investigated in the future.

## G. Derivation Details

In this section, we provide a more detailed derivation procedure about the minimization of the loss function Eq. (3).

**Update $\mathbf{A}_p$:** When updating $\mathbf{A}_p$, Eq. (3) equivalently becomes

$$\min_{\mathbf{A}_p} \sum_{p=1}^v \alpha_p^2 \|\mathbf{X}_p - \mathbf{A}_p\mathbf{T}_p\mathbf{B}_p\mathbf{C}\|_F^2 + \lambda \|\mathbf{A}_p\mathbf{T}_p - \mathbf{A}_p\mathbf{T}_p\mathbf{S}_p\|_F^2. \tag{17}$$

Due to the independence of views, anchor sets on different views are also independent of each other. Accordingly, we can equivalently transform Eq. (17) as

$$\min_{\mathbf{A}_p} \alpha_p^2 \|\mathbf{X}_p - \mathbf{A}_p\mathbf{T}_p\mathbf{B}_p\mathbf{C}\|_F^2 + \lambda \|\mathbf{A}_p\mathbf{T}_p - \mathbf{A}_p\mathbf{T}_p\mathbf{S}_p\|_F^2.$$

This is an unconstrained optimization problem, and according to the derivative value of zero, we can obtain

$$\begin{aligned}
\alpha_v^2 \left(\mathbf{A}_p\mathbf{T}_p\mathbf{B}_p\mathbf{C} - \mathbf{X}_p\right)\left(\mathbf{C}^\top\mathbf{B}_p^\top\mathbf{T}_p^\top\right) + \lambda\left(\mathbf{A}_p\mathbf{T}_p - \mathbf{A}_p\mathbf{T}_p\mathbf{S}_p\right)\left(\mathbf{T}_p^\top - \mathbf{S}_p^\top\mathbf{T}_p^\top\right) = 0 \\
\Rightarrow \alpha_v^2\mathbf{A}_p\mathbf{E}_p\mathbf{E}_p^\top - \alpha_v^2\mathbf{X}_p\mathbf{E}_p^\top + \lambda\mathbf{A}_p\mathbf{F}_p\mathbf{F}_p^\top = 0 \\
\Rightarrow \mathbf{A}_p\left(\alpha_v^2\mathbf{E}_p\mathbf{E}_p^\top + \lambda\mathbf{F}_p\mathbf{F}_p^\top\right) = \alpha_v^2\mathbf{X}_p\mathbf{E}_p^\top,
\end{aligned} \tag{18}$$

where $\mathbf{E}_p \in \mathbb{R}^{m \times n} = \mathbf{T}_p \mathbf{B}_p \mathbf{C}$, $\mathbf{F}_p \in \mathbb{R}^{m \times m} = \mathbf{T}_p - \mathbf{T}_p \mathbf{S}_p$. $\mathbf{T}_p$ is a permutation matrix, and thus is invertible. Further, according to the property of permutation matrix that its inverse is equal to its transposition, i.e., $\mathbf{T}_p^{-1} = \mathbf{T}_p^\top$, we have that $\mathbf{T}_p^{-1}$ is also a permutation matrix, and consequently can be seen as a series of elementary transformation operations. Based on the fact that elementary transformation does not change the rank of matrix, we have $rank(\mathbf{T}_p^{-1} \mathbf{F}_p) = rank(\mathbf{F}_p)$. Additionally, $rank(\mathbf{T}_p^{-1} \mathbf{F}_p) = rank(\mathbf{I} - \mathbf{S}_p)$. Since $\mathbf{S}_p$ is an anchor self-expression matrix and its diagonal elements are zero, we have $rank(\mathbf{I} - \mathbf{S}_p) = m$. That is, its rank is full. Thus, we have $rank(\mathbf{F}_p) = m$. It is full rank and accordingly is invertible. So, $\mathbf{F}_p \mathbf{F}_p^\top$ is also invertible. Further, the eigenvalue of $\mathbf{F}_p \mathbf{F}_p^\top$ is greater than 0, the eigenvalue of $\mathbf{E}_p \mathbf{E}_p^\top$ is greater than or equal to 0, and thus the eigenvalue of $(\boldsymbol{\alpha}_v^2 \mathbf{E}_p \mathbf{E}_p^\top + \lambda \mathbf{F}_p \mathbf{F}_p^\top)$ is greater than 0. Consequently, the item $\boldsymbol{\alpha}_v^2 \mathbf{E}_p \mathbf{E}_p^\top + \lambda \mathbf{F}_p \mathbf{F}_p^\top$ is invertible. Based on th above analysis, we can get that $\mathbf{A}_p = \boldsymbol{\alpha}_v^2 \mathbf{X}_p \mathbf{E}_p^\top \left( \boldsymbol{\alpha}_v^2 \mathbf{E}_p \mathbf{E}_p^\top + \lambda \mathbf{F}_p \mathbf{F}_p^\top \right)^{-1}$.

**Update $\mathbf{T}_p$:** When updating $\mathbf{T}_p$, Eq. (3) equivalently becomes

$$\min_{\mathbf{T}_p} \sum_{p=1}^{v} \boldsymbol{\alpha}_p^2 \|\mathbf{X}_p - \mathbf{A}_p \mathbf{T}_p \mathbf{B}_p \mathbf{C}\|_F^2 + \lambda \|\mathbf{A}_p \mathbf{T}_p - \mathbf{A}_p \mathbf{T}_p \mathbf{S}_p\|_F^2$$
$$\text{s.t. } \mathbf{T}_p^\top \mathbf{1} = \mathbf{1}, \mathbf{T}_p \mathbf{1} = \mathbf{1}, \mathbf{T}_p \in \{0,1\}^{m \times m}. \tag{19}$$

Due to $\mathbf{T}_p$ being performed on respective view, we can separately optimize each $\mathbf{T}_p$. Thus, Eq. (19) can be equivalently written as

$$\min_{\mathbf{T}_p} \boldsymbol{\alpha}_p^2 \|\mathbf{X}_p - \mathbf{A}_p \mathbf{T}_p \mathbf{B}_p \mathbf{C}\|_F^2 + \lambda \|\mathbf{A}_p \mathbf{T}_p - \mathbf{A}_p \mathbf{T}_p \mathbf{S}_p\|_F^2$$
$$\text{s.t. } \mathbf{T}_p^\top \mathbf{1} = \mathbf{1}, \mathbf{T}_p \mathbf{1} = \mathbf{1}, \mathbf{T}_p \in \{0,1\}^{m \times m}.$$

After expanding the objective using the trace operation and deleting irrelevant items, we can get

$$\min_{\mathbf{T}_p} \boldsymbol{\alpha}_p^2 \|\mathbf{X}_p - \mathbf{A}_p \mathbf{T}_p \mathbf{B}_p \mathbf{C}\|_F^2 + \lambda \|\mathbf{A}_p \mathbf{T}_p - \mathbf{A}_p \mathbf{T}_p \mathbf{S}_p\|_F^2$$
$$\Rightarrow \min_{\mathbf{T}_p} \text{Tr} \left( \boldsymbol{\alpha}_p^2 \mathbf{A}_p \mathbf{T}_p \mathbf{B}_p \mathbf{C} \mathbf{C}^\top \mathbf{B}_p^\top \mathbf{T}_p^\top \mathbf{A}_p^\top - 2\boldsymbol{\alpha}_p^2 \mathbf{A}_p^\top \mathbf{X}_p \mathbf{C}^\top \mathbf{B}_p^\top \mathbf{T}_p^\top + \lambda \mathbf{A}_p \mathbf{T}_p \mathbf{T}_p^\top \mathbf{A}_p^\top + \right. \tag{20}$$
$$\left. \lambda \mathbf{A}_p \mathbf{T}_p \mathbf{S}_p \mathbf{S}_p^\top \mathbf{T}_p^\top \mathbf{A}_p^\top - 2\lambda \mathbf{A}_p \mathbf{T}_p \mathbf{S}_p^\top \mathbf{T}_p^\top \mathbf{A}_p^\top \right).$$

According to the fact that $\mathbf{T}_p$ is a permutation matrix, we have $\mathbf{T}_p \mathbf{T}_p^\top = \mathbf{I}$. Additionally, considering that $\sum_{i=1}^{k} \mathbf{C}_{i,j} = 1, j \in \{1, 2, \ldots, n\}, \mathbf{C} \in \{0,1\}^{k \times n}$, we have that $\mathbf{C}\mathbf{C}^\top$ is a diagonal matrix, and its diagonal elements are $\sum_{j=1}^{n} \mathbf{C}_{i,j}$, $i = 1, 2, \cdots k$. Further, combined with $\mathbf{B}_p$ being orthogonal, we can obtain $\text{Tr} \left( \mathbf{B}_p \mathbf{C}\mathbf{C}^\top \mathbf{B}_p^\top \right) = \text{Tr} \left( \mathbf{C}\mathbf{C}^\top \right) = \sum_{i,j} \mathbf{C}_{i,j}$. Based on these analysis, we can get

$$\min_{\mathbf{T}_p} \boldsymbol{\alpha}_p^2 \|\mathbf{X}_p - \mathbf{A}_p \mathbf{T}_p \mathbf{B}_p \mathbf{C}\|_F^2 + \lambda \|\mathbf{A}_p \mathbf{T}_p - \mathbf{A}_p \mathbf{T}_p \mathbf{S}_p\|_F^2$$
$$\Rightarrow \min_{\mathbf{T}_p} \text{Tr} \left( -2\boldsymbol{\alpha}_p^2 \mathbf{T}_p^\top \mathbf{A}_p^\top \mathbf{X}_p \mathbf{C}^\top \mathbf{B}_p^\top + \lambda \mathbf{T}_p^\top \mathbf{A}_p^\top \mathbf{A}_p \mathbf{T}_p \mathbf{S}_p \mathbf{S}_p^\top + \right.$$
$$\left. \boldsymbol{\alpha}_p^2 \mathbf{T}_p^\top \mathbf{A}_p^\top \mathbf{A}_p \mathbf{T}_p \mathbf{B}_p \mathbf{C}\mathbf{C}^\top \mathbf{B}_p^\top - 2\lambda \mathbf{T}_p^\top \mathbf{A}_p^\top \mathbf{A}_p \mathbf{T}_p \mathbf{S}_p^\top \right) \tag{21}$$
$$\Rightarrow \min_{\mathbf{T}_p} \text{Tr} \left( \lambda \mathbf{T}_p^\top \mathbf{G}_p \mathbf{T}_p \mathbf{H}_p + \boldsymbol{\alpha}_p^2 \mathbf{T}_p^\top \mathbf{G}_p \mathbf{T}_p \mathbf{M}_p - 2\lambda \mathbf{T}_p^\top \mathbf{G}_p \mathbf{T}_p \mathbf{S}_p^\top - 2\boldsymbol{\alpha}_p^2 \mathbf{T}_p^\top \mathbf{J}_p \right)$$
$$\Rightarrow \min_{\mathbf{T}_p} \text{Tr} \left( \mathbf{T}_p^\top \mathbf{G}_p \mathbf{T}_p \left( \lambda \mathbf{H}_p + \boldsymbol{\alpha}_p^2 \mathbf{M}_p \right) - 2\lambda \mathbf{T}_p^\top \mathbf{G}_p \mathbf{T}_p \mathbf{S}_p^\top - 2\boldsymbol{\alpha}_p^2 \mathbf{T}_p^\top \mathbf{J}_p \right),$$

where $\mathbf{G}_p \in \mathbb{R}^{m \times m} = \mathbf{A}_p^\top \mathbf{A}_p$, $\mathbf{H}_p \in \mathbb{R}^{m \times m} = \mathbf{S}_p \mathbf{S}_p^\top$, $\mathbf{M}_p \in \mathbb{R}^{m \times m} = \mathbf{B}_p \mathbf{C}\mathbf{C}^\top \mathbf{B}_p^\top$, $\mathbf{J}_p \in \mathbb{R}^{m \times m} = \mathbf{A}_p^\top \mathbf{X}_p \mathbf{C}^\top \mathbf{B}_p^\top$. Combined with the feasible region in Eq. (19), we can determine the optimal solution of $\mathbf{T}_p$ via traversal searching using $[\mathbf{e}_1, \mathbf{e}_2, \cdots, \mathbf{e}_i, \cdots, \mathbf{e}_m]$ where $\mathbf{e}_i$ is the one-hot vector. Kindly note that the size of $\mathbf{T}_p$ is $m \times m$ and $m$ is generally small, performing traversal searching on $\mathbf{T}_p$ will not incur significant computing costs.

**Update $\mathbf{B}_p$:** When updating $\mathbf{B}_p$, Eq. (3) equivalently becomes

$$\min_{\mathbf{B}_p} \sum_{p=1}^{v} \boldsymbol{\alpha}_p^2 \|\mathbf{X}_p - \mathbf{A}_p \mathbf{T}_p \mathbf{B}_p \mathbf{C}\|_F^2 + \beta \text{Tr}(\mathbf{B}_p^\top \mathbf{L_s} \mathbf{B}_p \mathbf{C}\mathbf{C}^\top)$$
$$\text{s.t. } \mathbf{B}_p^\top \mathbf{B}_p = \mathbf{I}_k. \tag{22}$$

Since the basic coefficient matrices $\{\mathbf{B}_p\}_{p=1}^{v}$ on different views are independent of each other, we can equivalently transform Eq. (22) as

$$\min_{\mathbf{B}_p} \boldsymbol{\alpha}_p^2 \|\mathbf{X}_p - \mathbf{A}_p\mathbf{T}_p\mathbf{B}_p\mathbf{C}\|_F^2 + \beta \operatorname{Tr}(\mathbf{B}_p^\top \mathbf{L_s}\mathbf{B}_p\mathbf{C}\mathbf{C}^\top)$$

$$\text{s.t. } \mathbf{B}_p^\top \mathbf{B}_p = \mathbf{I}_k. \tag{23}$$

Expanding the objective and then deleting irrelevant items, we can obtain

$$\min_{\mathbf{B}_p} \boldsymbol{\alpha}_p^2 \|\mathbf{X}_p - \mathbf{A}_p\mathbf{T}_p\mathbf{B}_p\mathbf{C}\|_F^2 + \beta \operatorname{Tr}(\mathbf{B}_p^\top \mathbf{L_s}\mathbf{B}_p\mathbf{C}\mathbf{C}^\top)$$

$$\Rightarrow \min_{\mathbf{B}_p} \operatorname{Tr}\left(\boldsymbol{\alpha}_p^2 \mathbf{A}_p\mathbf{T}_p\mathbf{B}_p\mathbf{C}\mathbf{C}^\top\mathbf{B}_p^\top\mathbf{T}_p^\top\mathbf{A}_p^\top - 2\boldsymbol{\alpha}_p^2\mathbf{T}_p^\top\mathbf{A}_p^\top\mathbf{X}_p\mathbf{C}^\top\mathbf{B}_p^\top + \beta\mathbf{B}_p^\top \mathbf{L_s}\mathbf{B}_p\mathbf{C}\mathbf{C}^\top\right) \tag{24}$$

Since $\mathbf{C}\mathbf{C}^\top$ is diagonal and $\mathbf{B}_p$ is orthogonal, we can further have

$$\min_{\mathbf{B}_p} \boldsymbol{\alpha}_p^2 \|\mathbf{X}_p - \mathbf{A}_p\mathbf{T}_p\mathbf{B}_p\mathbf{C}\|_F^2 + \beta \operatorname{Tr}(\mathbf{B}_p^\top \mathbf{L_s}\mathbf{B}_p\mathbf{C}\mathbf{C}^\top)$$

$$\Rightarrow \min_{\mathbf{B}_p} \operatorname{Tr}\left(\beta\mathbf{B}_p^\top \mathbf{L_s}\mathbf{B}_p\mathbf{C}\mathbf{C}^\top + \boldsymbol{\alpha}_p^2\mathbf{B}_p^\top\mathbf{Q}_p\mathbf{B}_p\mathbf{C}\mathbf{C}^\top - 2\boldsymbol{\alpha}_p^2\mathbf{T}_p^\top\mathbf{A}_p^\top\mathbf{X}_p\mathbf{C}^\top\mathbf{B}_p^\top\right) \tag{25}$$

$$\Rightarrow \min_{\mathbf{B}_p} \operatorname{Tr}\left(\mathbf{B}_p^\top \left(\beta\mathbf{L_s} + \boldsymbol{\alpha}_p^2\mathbf{Q}_p\right)\mathbf{B}_p\mathbf{C}\mathbf{C}^\top - 2\boldsymbol{\alpha}_p^2\mathbf{C}\mathbf{X}_p^\top\mathbf{A}_p\mathbf{T}_p\mathbf{B}_p\right),$$

where $\mathbf{Q}_p \in \mathbb{R}^{m \times m} = \mathbf{T}_p^\top \mathbf{A}_p^\top \mathbf{A}_p \mathbf{T}_p$.

Considering that the feasible region $\mathbf{B}_p^\top\mathbf{B}_p = \mathbf{I}_k$ can be equivalently divided into $[\mathbf{B}_p]_{:,j}^\top[\mathbf{B}_p]_{:,j} = 1$ and $[\mathbf{B}_p]_{:,j}^\top[\mathbf{B}_p]_{:,i} = 0, i = 1, 2, \cdots, k, i \neq j, j = 1, 2, \cdots, k$, we can solve $\mathbf{B}_p$ column by column. Thus, we have

$$\min_{\mathbf{B}_p} \operatorname{Tr}\left(\mathbf{B}_p^\top \left(\beta\mathbf{L_s} + \boldsymbol{\alpha}_p^2\mathbf{Q}_p\right)\mathbf{B}_p\mathbf{C}\mathbf{C}^\top - 2\boldsymbol{\alpha}_p^2\mathbf{C}\mathbf{X}_p^\top\mathbf{A}_p\mathbf{T}_p\mathbf{B}_p\right)$$

$$\Rightarrow \min_{[\mathbf{B}_p]_{:,j}} [\mathbf{B}_p^\top]_{j,:} \left(\beta\mathbf{L_s} + \boldsymbol{\alpha}_p^2\mathbf{Q}_p\right)\mathbf{B}_p[\mathbf{C}\mathbf{C}^\top]_{:,j} + \left[-2\boldsymbol{\alpha}_p^2\mathbf{C}\mathbf{X}_p^\top\mathbf{A}_p\mathbf{T}_p\right]_{j,:}[\mathbf{B}_p]_{:,j} \tag{26}$$

$$\Rightarrow \min_{[\mathbf{B}_p]_{:,j}} [\mathbf{B}_p]_{:,j}^\top \sum_{i=1}^{n}\mathbf{C}_{j,i}\left(\beta\mathbf{L_s} + \boldsymbol{\alpha}_p^2\mathbf{Q}_p\right)[\mathbf{B}_p]_{:,j} + \left[-2\boldsymbol{\alpha}_p^2\mathbf{C}\mathbf{X}_p^\top\mathbf{A}_p\mathbf{T}_p\right]_{j,:}[\mathbf{B}_p]_{:,j},$$

where the objective is quadratic. Besides, the constraint $[\mathbf{B}_p]_{:,j}^\top[\mathbf{B}_p]_{:,j} = 1$ can be equivalently written as $[\mathbf{B}_p]_{:,j}^\top\mathbf{I}_{m \times m}[\mathbf{B}_p]_{:,j} - 1 = 0$. $[\mathbf{B}_p]_{:,j}^\top[\mathbf{B}_p]_{:,i} = 0, i = 1, 2, \cdots, k, i \neq j$ can be written as $[[\mathbf{B}_p]_{:,1}, [\mathbf{B}_p]_{:,2}, \cdots, [\mathbf{B}_p]_{:,j-1}, [\mathbf{B}_p]_{:,j+1}, \cdots, [\mathbf{B}_p]_{:,k}]^\top[\mathbf{B}_p]_{:,j} = \mathbf{0}_{(k-1) \times 1}$. Apparently, the constraints are also quadratic. Consequently, the optimization problem about $\mathbf{B}_p$ can be equivalently transformed as

$$\min_{[\mathbf{B}_p]_{:,j}} [\mathbf{B}_p]_{:,j}^\top \sum_{i=1}^{n}\mathbf{C}_{j,i}\left(\beta\mathbf{L_s} + \boldsymbol{\alpha}_p^2\mathbf{Q}_p\right)[\mathbf{B}_p]_{:,j} + \left[-2\boldsymbol{\alpha}_p^2\mathbf{C}\mathbf{X}_p^\top\mathbf{A}_p\mathbf{T}_p\right]_{j,:}[\mathbf{B}_p]_{:,j}$$

$$\text{s.t. } [\mathbf{B}_p]_{:,j}^\top\mathbf{I}_{m \times m}[\mathbf{B}_p]_{:,j} - 1 = 0,$$

$$[[\mathbf{B}_p]_{:,1}, [\mathbf{B}_p]_{:,2}, \cdots, [\mathbf{B}_p]_{:,j-1}, [\mathbf{B}_p]_{:,j+1}, \cdots, [\mathbf{B}_p]_{:,k}]^\top[\mathbf{B}_p]_{:,j} = \mathbf{0}_{(k-1) \times 1}.$$

This is a QCQP optimization problem, and can be solved in $\mathcal{O}(m^3)$ computing complexity.

**Update** $\mathbf{S}_p$: When updating $\mathbf{S}_p$, Eq. (3) equivalently becomes

$$\min_{\mathbf{S}_p} \lambda \|\mathbf{A}_p\mathbf{T}_p - \mathbf{A}_p\mathbf{T}_p\mathbf{S}_p\|_F^2 + \beta \operatorname{Tr}(\mathbf{B}_p^\top \mathbf{L_s}\mathbf{B}_p\mathbf{C}\mathbf{C}^\top)$$

$$\text{s.t. } \mathbf{S}_p^\top \mathbf{1} = \mathbf{1}, \mathbf{S}_p \geq 0, \sum_{i=1}^{m}[\mathbf{S}_p]_{i,i} = 0. \tag{27}$$

Expanding the objective, we have

$$
\min_{\mathbf{S}_p} \lambda \left\| \mathbf{A}_p \mathbf{T}_p - \mathbf{A}_p \mathbf{T}_p \mathbf{S}_p \right\|_F^2 + \beta \operatorname{Tr}(\mathbf{B}_p^\top \mathbf{L_s} \mathbf{B}_p \mathbf{C} \mathbf{C}^\top)
$$

$$
\Rightarrow \min_{\mathbf{S}_p} \operatorname{Tr} \left( \lambda \mathbf{A}_p \mathbf{T}_p \mathbf{S}_p \mathbf{S}_p^\top \mathbf{T}_p^\top \mathbf{A}_p^\top - 2\lambda \mathbf{A}_p \mathbf{T}_p \mathbf{S}_p^\top \mathbf{T}_p^\top \mathbf{A}_p^\top \right.
$$

$$
\left. + \lambda \mathbf{A}_p \mathbf{T}_p \mathbf{T}_p^\top \mathbf{A}_p^\top + \beta \mathbf{B}_p^\top \mathbf{D}_p \mathbf{B}_p \mathbf{C} \mathbf{C}^\top - \beta \mathbf{B}_p^\top \mathbf{S}_p \mathbf{B}_p \mathbf{C} \mathbf{C}^\top \right)
$$

$$
\Rightarrow \min_{\mathbf{S}_p} \operatorname{Tr} \left( \lambda \mathbf{A}_p \mathbf{T}_p \mathbf{S}_p \mathbf{S}_p^\top \mathbf{T}_p^\top \mathbf{A}_p^\top - 2\lambda \mathbf{A}_p \mathbf{T}_p \mathbf{S}_p^\top \mathbf{T}_p^\top \mathbf{A}_p^\top - \beta \mathbf{B}_p^\top \mathbf{S}_p \mathbf{B}_p \mathbf{C} \mathbf{C}^\top \right) \tag{28}
$$

$$
\Rightarrow \min_{\mathbf{S}_p} \operatorname{Tr} \left( \lambda \mathbf{S}_p^\top \mathbf{T}_p^\top \mathbf{A}_p^\top \mathbf{A}_p \mathbf{T}_p \mathbf{S}_p - 2\lambda \mathbf{T}_p^\top \mathbf{A}_p^\top \mathbf{A}_p \mathbf{T}_p \mathbf{S}_p - \beta \mathbf{B}_p \mathbf{C} \mathbf{C}^\top \mathbf{B}_p^\top \mathbf{S}_p \right)
$$

$$
\Rightarrow \min_{\mathbf{S}_p} \operatorname{Tr} \left( \lambda \mathbf{S}_p^\top \mathbf{Q}_p \mathbf{S}_p - 2\lambda \mathbf{Q}_p \mathbf{S}_p - \beta \mathbf{M}_p \mathbf{S}_p \right)
$$

$$
\Rightarrow \min_{\mathbf{S}_p} \operatorname{Tr} \left( \mathbf{S}_p^\top \mathbf{Q}_p \mathbf{S}_p + 2 \left( -\mathbf{Q}_p - \frac{\beta}{2\lambda} \mathbf{M}_p \right) \mathbf{S}_p \right),
$$

where $\mathbf{Q}_p \in \mathbb{R}^{m \times m} = \mathbf{T}_p^\top \mathbf{A}_p^\top \mathbf{A}_p \mathbf{T}_p$, $\mathbf{M}_p \in \mathbb{R}^{m \times m} = \mathbf{B}_p \mathbf{C} \mathbf{C}^\top \mathbf{B}_p^\top$.

Noticed that the feasible region is for each column of $\mathbf{S}_p$, consequently, we can equivalently rewrite the constraints in the form of columns. That is, we can transform $\mathbf{S}_p^\top \mathbf{1} = \mathbf{1}, \mathbf{S}_p \geq 0, \sum_{i=1}^m [\mathbf{S}_p]_{i,i} = 0$ as $[\mathbf{S}_p]_{:,j}^\top \mathbf{1} = 1, [\mathbf{S}_p]_{:,j} \geq 0, [\mathbf{S}_p]_{j,j} = 0, j = 1, 2, \cdots, m$. Further, we can transform $[\mathbf{S}_p]_{j,j} = 0, j = 1, 2, \cdots, m$ as $\mathbf{e}_j^\top [\mathbf{S}_p]_{:,j} = 0, j = 1, 2, \cdots, m$, where $\mathbf{e}_j$ is the one-hot vector.

Based on these, for the objective function, we can further have

$$
\min_{\mathbf{S}_p} \operatorname{Tr} \left( \mathbf{S}_p^\top \mathbf{Q}_p \mathbf{S}_p + 2 \left( -\mathbf{Q}_p - \frac{\beta}{2\lambda} \mathbf{M}_p \right) \mathbf{S}_p \right)
$$

$$
\Rightarrow \min_{[\mathbf{S}_p]_{:,j}} \frac{1}{2} [\mathbf{S}_p]_{:,j}^\top \mathbf{Q}_p [\mathbf{S}_p]_{:,j} + \left( -\mathbf{Q}_p - \frac{\beta}{2\lambda} \mathbf{M}_p \right)_{j,:} [\mathbf{S}_p]_{:,j}. \tag{29}
$$

Therefore, the optimization problem about $\mathbf{S}_p$ can be equivalently written as

$$
\min_{[\mathbf{S}_p]_{:,j}} \frac{1}{2} [\mathbf{S}_p]_{:,j}^\top \mathbf{Q}_p [\mathbf{S}_p]_{:,j} + \left( -\mathbf{Q}_p - \frac{\beta}{2\lambda} \mathbf{M}_p \right)_{j,:} [\mathbf{S}_p]_{:,j}
$$
$$
\text{s.t. } [\mathbf{S}_p]_{:,j}^\top \mathbf{1} = 1, 0 \leq [\mathbf{S}_p]_{:,j}, \mathbf{e}_j^\top [\mathbf{S}_p]_{:,j} = 0, j = 1, 2, \cdots, m, \tag{30}
$$

which is a QP problem, and can be solved within $\mathcal{O}(m^2)$ computing complexity.

**Update** $\mathbf{C}$: When updating $\mathbf{C}$, Eq. (3) equivalently becomes

$$
\min_{\mathbf{C}} \sum_{p=1}^v \alpha_p^2 \left\| \mathbf{X}_p - \mathbf{A}_p \mathbf{T}_p \mathbf{B}_p \mathbf{C} \right\|_F^2 + \beta \operatorname{Tr}(\mathbf{B}_p^\top \mathbf{L_s} \mathbf{B}_p \mathbf{C} \mathbf{C}^\top)
$$
$$
\text{s.t. } \sum_{i=1}^k \mathbf{C}_{i,j} = 1, j = 1, 2, \ldots, n, \mathbf{C} \in \{0, 1\}^{k \times n}. \tag{31}
$$

For the objective function, we have

$$
\min_{\mathbf{C}} \sum_{p=1}^v \alpha_p^2 \left\| \mathbf{X}_p - \mathbf{A}_p \mathbf{T}_p \mathbf{B}_p \mathbf{C} \right\|_F^2 + \beta \operatorname{Tr}(\mathbf{B}_p^\top \mathbf{L_s} \mathbf{B}_p \mathbf{C} \mathbf{C}^\top)
$$

$$
\Rightarrow \min_{\mathbf{C}} \operatorname{Tr} \left( \mathbf{C}^\top \sum_{p=1}^v \alpha_p^2 \mathbf{B}_p^\top \mathbf{T}_p^\top \mathbf{A}_p^\top \mathbf{A}_p \mathbf{T}_p \mathbf{B}_p \mathbf{C} - 2 \sum_{p=1}^v \alpha_p^2 \mathbf{X}_p^\top \mathbf{A}_p \mathbf{T}_p \mathbf{B}_p \mathbf{C} + \beta \mathbf{C}^\top \sum_{p=1}^v \mathbf{B}_p^\top \mathbf{L_s} \mathbf{B}_p \mathbf{C} \right) \tag{32}
$$

$$
\Rightarrow \min_{\mathbf{C}} \operatorname{Tr} \left( \mathbf{C}^\top \left( \sum_{p=1}^v \alpha_p^2 \mathbf{B}_p^\top \mathbf{T}_p^\top \mathbf{A}_p^\top \mathbf{A}_p \mathbf{T}_p \mathbf{B}_p + \beta \mathbf{B}_p^\top \mathbf{L_s} \mathbf{B}_p \right) \mathbf{C} - 2 \sum_{p=1}^v \alpha_p^2 \mathbf{X}_p^\top \mathbf{A}_p \mathbf{T}_p \mathbf{B}_p \mathbf{C} \right)
$$

$$
\Rightarrow \min_{\mathbf{C}} \operatorname{Tr} \left( \mathbf{C}^\top \mathbf{W} \mathbf{C} - \mathbf{Z} \mathbf{C} \right),
$$

where $\mathbf{W} \in \mathbb{R}^{k \times k} = \sum_{p=1}^{v} \alpha_p^2 \mathbf{B}_p^\top \mathbf{T}_p^\top \mathbf{A}_p^\top \mathbf{A}_p \mathbf{T}_p \mathbf{B}_p + \beta \mathbf{B}_p^\top \mathbf{L_s} \mathbf{B}_p$, $\mathbf{Z} \in \mathbb{R}^{n \times k} = 2 \sum_{p=1}^{v} \alpha_p^2 \mathbf{X}_p^\top \mathbf{A}_p \mathbf{T}_p \mathbf{B}_p$.

The constraints mean that there is only one non-zero element in each column of $\mathbf{C}$, and consequently we can optimize $\mathbf{C}$ by column. We can get

$$\min_{\mathbf{C}} \operatorname{Tr} \left( \mathbf{C}^\top \mathbf{W} \mathbf{C} - \mathbf{Z} \mathbf{C} \right) \Rightarrow \min_{\mathbf{C}_{:,j}} \mathbf{C}_{:,j}^\top \mathbf{W} \mathbf{C}_{:,j} - \mathbf{Z}_{j,:} \mathbf{C}_{:,j}. \tag{33}$$

Further, the item $\mathbf{C}_{:,j}^\top \mathbf{W} \mathbf{C}_{:,j}$ indicates that it takes a diagonal element of $\mathbf{W}$, and $\mathbf{Z}_{j,:} \mathbf{C}_{:,j}$ indicates that it takes a element of $\mathbf{Z}_{j,:}$. Thus, we can determine the corresponding index of minimum by

$$l^* = \arg \min_{l} \mathbf{W}_{l,l} - \mathbf{Z}_{j,l}, \quad l = 1, 2, \cdots, k. \tag{34}$$

Then, the value of $\mathbf{C}_{:,j}$ can be determined by assigning $\mathbf{C}_{l^*,j}$ as 1 while assigning other elements of $\mathbf{C}_{:,j}$ as 0.

**Update $\alpha$**: When updating $\alpha$, Eq. (3) equivalently becomes

$$\min_{\alpha} \sum_{p=1}^{v} \alpha_p^2 \| \mathbf{X}_p - \mathbf{A}_p \mathbf{T}_p \mathbf{B}_p \mathbf{C} \|_F^2$$
$$\text{s.t. } \alpha^\top \mathbf{1} = 1, \alpha \geq 0.$$

Considering that the term $\frac{1}{b_p} = \| \mathbf{X}_p - \mathbf{A}_p \mathbf{T}_p \mathbf{B}_p \mathbf{C} \|_F^2$ is a constant with respect to $\alpha$, we can solve $\alpha$ using Cauchy inequality. Specially, we can get that the optimal solution is $\alpha_p = \frac{b_p}{\sum_{p=1}^{v} b_p}$.

## H. More Conclusions for Table 1

1. On CALTE7, MFLVC receives better clustering results in NMI, probably because it achieves reconstruction and consistency by learning features at multiple levels rather than at single level for each view, and utilizes the consensus semantics shared in all views and semantic labels to decrease the view-private unfavorable influence.

2. FPMVS achieves 0.29% increasement in terms of ACC on CIF10Tra4, and possible reasons are that it employs a group of projectors to maintain the anchor dimension consistency and extracts consensus multi-view isomeric features by utilizing an unified graph structure with cluster distribution constraints.

3. On Cora in Fscore, MSCIAS slightly surpasses us with 0.72%, which is mainly because it enforces encoded similarity to maximally depend on the potential intact-samples through HSIC criterion and utilizes the local connectivity of intact space to eliminate outliers and enhance the distinguishability of similarity.

4. For SFMC, it makes preferable results on REU7200 in NMI, main reasons of which could be that it integrates connectivity constraint into the learning of joint graph to reflect cluster distribution and adaptively adjusts the graph contributions on different views in self-supervised weighting way.

5. PMSC, AMGL and MLRSSC express inferior performance in certain scenarios, the reasons of which could be that PMSC reaches the consensus clustering under the premise of the basic partition realizing the ground truth and meanwhile equally treats every view, the factors generated by the cluster indicator with orthogonal constraints in AMGL impair the discriminability of some graphs, and MLRSSC linearly combines the generated representation matrices and only utilizes truncation operation to determine the penalty parameters.

## I. Other Ablation Studies

In the paper, rather than treating views equally, we adopt a square weighting scheme to adaptively combine views together. To validate the effectiveness of this strategy, we conduct the comparison experiments with equal view weighting (EVW). The results are summarized in Table 9, where AVW denotes the clustering results based on our adaptive view weighting. Obviously, AVW receives more desirable results than EVW in most cases, which suggests that the adaptive view weighting strategy is recommendable. Additionally, we also plot the learned view weights, as shown in Fig. 3. It can be seen that it indeed assigns different weights to measure the contribution between views.

Table 9. The effectiveness of view weighting

| Metric | Ablation | DERMATO | CALTE7 | Cora | REU7200 | Reuters | CIF10Tra4 | FasMNI4V |
|--------|----------|---------|--------|------|---------|---------|-----------|----------|
| ACC | EVW | 84.59 | 76.73 | 44.94 | 16.82 | 47.84 | 25.26 | 52.69 |
| | AVW | **85.47** | **80.66** | **52.44** | **26.22** | **54.26** | **26.83** | **57.36** |
| NMI | EVW | 89.44 | 42.88 | 33.06 | 3.60 | 29.54 | 15.20 | 56.84 |
| | AVW | **89.97** | **45.25** | **43.70** | **6.25** | **31.87** | **15.64** | **59.21** |
| Fscore | EVW | 86.59 | 71.80 | 35.36 | **28.77** | 42.32 | 18.03 | 46.90 |
| | AVW | **87.92** | **78.12** | **41.12** | 28.55 | **44.84** | **20.64** | **51.37** |

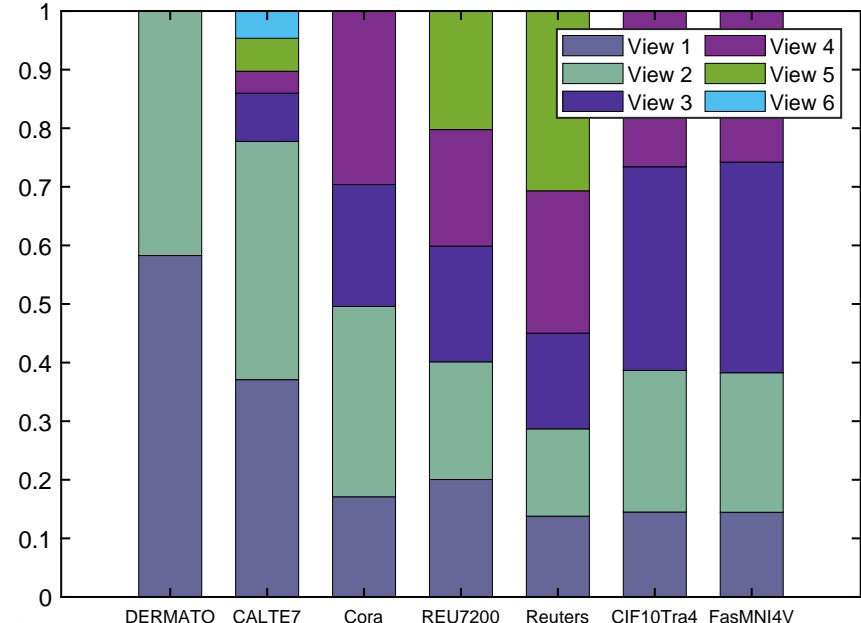

Figure 3. The learned view weights on seven public datasets.

Besides, unlike current techniques generating anchors via random sampling or heuristic searching, which leads to anchors being separated from subsequent procedures like graph learning and spectrum construction, we integrate anchors into objective optimization framework to make them able to interact with other parts and thereby facilitate each other. To investigate its effectiveness, we organize corresponding ablation experiments and present the comparison results in Table 10, where HS denotes the clustering results based on anchors generated by heuristic searching while LA denotes the results based on our anchor learning. It is easy to observe that LA outperforms HS with noticeable margins, which illustrates that the anchor learning strategy is functional and can provide more pleasing clustering results.

Table 10. The effectiveness of learnable anchor

| Metric | Ablation | DERMATO | CALTE7 | Cora | REU7200 | Reuters | CIF10Tra4 | FasMNI4V |
|--------|----------|---------|--------|------|---------|---------|-----------|----------|
| ACC | HS | 65.64 | 64.59 | 30.24 | 16.68 | 27.20 | 24.08 | 47.21 |
| | LA | **85.47** | **80.66** | **52.44** | **26.22** | **54.26** | **26.83** | **57.36** |
| NMI | HS | 69.84 | 37.95 | 33.54 | 1.06 | 1.43 | 12.98 | 47.07 |
| | LA | **89.97** | **45.25** | **43.70** | **6.25** | **31.87** | **15.64** | **59.21** |
| Fscore | HS | 69.33 | 61.54 | 30.40 | 24.43 | 35.25 | 18.03 | 41.43 |
| | LA | **87.92** | **78.12** | **41.12** | **28.55** | **44.84** | **20.64** | **51.37** |

## J. More Illustrations for Table 7

The selection of baseline view not only brings complicated solving procedure but also affects the clustering performance. If the baseline view is not well selected, the graph structure will be inaccurately fused. Unlike this, we do not require the baseline view, and can automatically rearrange anchors according to respective view characteristics. Besides, we also can coordinate with anchors in the unified framework and thereby facilitate the learning of anchors, which makes view information interact across different levels.

## K. The Effectiveness of Anchor Self-expression

The self-expression affinity learning is utilized to construct the sample-sample affinity with full size in subspace clustering. Inspired by this, we explicitly extract the global structure between anchors via self-expression learning, and meanwhile feed that into anchor-sample so as to better exploit the manifold characteristics hidden within samples. (Kindly note that in this work, we did not calculate the **sample-sample** relations through self-expression learning.) In addition to this, our work also designs a joint-alignment mechanism which does not involve the selection of the baseline view and meanwhile can cooperate with the learning of anchors. Moreover, a solving scheme with linear complexity enables our framework to effectively tackle MVC tasks.

Anchor self-expression learning can help extract the geometric characteristics between anchors, and meanwhile facilities the learning of anchors owing to the joint-optimization mechanism. To validate this point, we organize four groups of comparison experiments, i.e., No self-expression + No leanring (NSNL), No self-expression + Having learning (NSHL), Having self-expression + No learning (HSNL), Having self-expression + Having learning (HSHL, i.e., Ours). The comparison results are reported in Table 11.

*Table 11.* The effectiveness of anchor self-expression

| Dataset | DERMATO | CALTE7 | Cora | REU7200 | Reuters | CIF10Tra4 | FasMNI4V |
|---------|---------|--------|------|---------|---------|-----------|----------|
| | | | | ACC(%) | | | |
| NSNL | 60.42 | 43.88 | 27.96 | 14.33 | 25.32 | 19.73 | 41.27 |
| NSHL | 71.51 | 49.05 | 30.35 | 16.75 | 47.05 | 26.69 | 52.15 |
| HSNL | 65.64 | 64.59 | 30.24 | 16.68 | 27.20 | 24.08 | 47.21 |
| HSHL | **85.47** | **80.66** | **52.44** | **26.22** | **54.26** | **26.83** | **57.36** |
| | | | | NMI(%) | | | |
| NSNL | 64.37 | 35.68 | 5.88 | 1.01 | 1.38 | 12.57 | 44.79 |
| NSHL | 83.97 | 40.21 | 6.02 | 2.53 | 23.19 | 15.48 | 58.13 |
| HSNL | 69.84 | 37.95 | 33.54 | 1.06 | 1.43 | 12.98 | 47.07 |
| HSHL | **89.97** | **45.25** | **43.70** | **6.25** | **31.87** | **15.64** | **59.21** |
| | | | | Fscore(%) | | | |
| NSNL | 63.76 | 48.43 | 28.79 | 23.42 | 33.87 | 16.86 | 37.64 |
| NSHL | 73.79 | 51.25 | 30.42 | 28.54 | 43.04 | 17.70 | 46.77 |
| HSNL | 69.33 | 61.54 | 30.40 | 24.43 | 35.25 | 18.03 | 41.43 |
| HSHL | **87.92** | **78.12** | **41.12** | **28.55** | **44.84** | **20.64** | **51.37** |

As seen, HSNL is consistently preferable than NSNL, and HSHL is consistently preferable than NSHL. These demonstrate that the anchor self-expression can facilitate the clustering performance improvement. Additionally, NSHL is consistently preferable than NSNL, and HSHL is consistently preferable than HSNL. These illustrate that the anchor learning can help increase the clustering results. Therefore, we can conclude that the anchor self-expression learning can enhance the quality of anchors to increase the clustering results.

## L. The Effectiveness of View-specific Anchors

The works based on feature space fusion or shared representations usually extract a group of unified anchors rather than multiple groups of view-specific anchors to construct the similarity relationship. Although avoiding alignment, this paradigm

could not effectively exploit complementary information between views due to the unified anchors being shared for all views. To further illustrate this point, we organize the comparison experiments between unified anchors (UA) and view-specific anchors (VSA, i.e., ours). The results are presented in Table 12.

*Table 12.* The effectiveness of view-specific anchors

| Dataset | DERMATO | CALTE7 | Cora | REU7200 | Reuters | CIF10Tra4 | FasMNI4V |
|---------|---------|--------|------|---------|---------|-----------|----------|
| ACC(%) | | | | | | | |
| UA | 81.23 | 71.33 | 49.73 | 24.97 | 48.46 | 22.36 | 53.38 |
| VSA | **85.47** | **80.66** | **52.44** | **26.22** | **54.26** | **26.83** | **57.36** |
| NMI(%) | | | | | | | |
| UA | 82.76 | 42.26 | 39.87 | 6.03 | 29.89 | 13.43 | 51.97 |
| VSA | **89.97** | **45.25** | **43.70** | **6.25** | **31.87** | **15.64** | **59.21** |
| Fscore(%) | | | | | | | |
| UA | 80.64 | 69.72 | **42.28** | 25.21 | **46.13** | 19.58 | **52.21** |
| VSA | **87.92** | **78.12** | 41.12 | **28.55** | 44.84 | **20.64** | 51.37 |

As seen, our results are more preferable than the counterparts adopting unified anchors, the reason of which could be that the view-exclusive complementary representation information (view-specific (aligned) anchors contain) outweighs the view-common consensus representation information (unified anchors contain).

*Table 13.* Single-view experimental results in ACC

| Dataset | DERMATO | CALTE7 | Cora | REU7200 | Reuters | CIF10Tra4 | FasMNI4V |
|---------|---------|--------|------|---------|---------|-----------|----------|
| FMR | 66.50($\pm$5.19) | 21.59($\pm$0.49) | 24.08($\pm$0.38) | - | - | - | - |
| PMSC | 64.83($\pm$8.99) | 38.18($\pm$1.81) | 27.93($\pm$0.86) | 19.24($\pm$0.53) | - | - | - |
| AMGL | 22.09($\pm$0.16) | 39.72($\pm$1.35) | 14.67($\pm$0.17) | 17.00($\pm$0.05) | 20.32($\pm$0.27) | - | - |
| MSCIAS | 64.35($\pm$8.27) | 40.63($\pm$2.93) | 30.88($\pm$1.30) | 19.25($\pm$0.93) | 25.38($\pm$1.44) | - | - |
| MVSC | 57.08($\pm$9.81) | 45.54($\pm$2.03) | - | - | - | - | - |
| MLRSSC | 31.01($\pm$0.00) | 50.14($\pm$0.00) | 30.21($\pm$0.00) | 16.92($\pm$0.00) | - | - | - |
| MPAC | 61.14($\pm$0.00) | 41.72($\pm$0.00) | 32.48($\pm$0.00) | 18.40($\pm$0.00) | - | - | - |
| MCLES | 64.47($\pm$4.27) | 46.33($\pm$2.58) | 30.47($\pm$1.32) | - | - | - | - |
| FMCNOF | 52.51($\pm$4.93) | 48.51($\pm$4.22) | 24.34($\pm$2.43) | 18.67($\pm$2.38) | - | 19.55($\pm$1.86) | 31.52($\pm$2.21) |
| ADAGAE | - | - | - | - | - | - | - |
| PFSC | 63.80($\pm$6.28) | 50.62($\pm$4.77) | - | - | - | - | - |
| SFMC | 65.88($\pm$0.00) | 50.24($\pm$0.00) | 30.17($\pm$0.00) | 15.75($\pm$0.00) | 20.53($\pm$0.00) | 22.08($\pm$0.00) | - |
| MSGL | 29.61($\pm$1.03) | - | - | 17.99($\pm$0.88) | 23.83($\pm$0.64) | 21.48($\pm$0.57) | - |
| DEMVC | - | - | - | - | - | - | - |
| FPMVS | **68.46($\pm$7.24)** | 49.88($\pm$2.19) | 32.05($\pm$1.91) | 19.55($\pm$0.19) | 22.90($\pm$2.15) | 22.37($\pm$0.62) | 51.97($\pm$3.26) |
| 3AMVC | - | - | - | - | - | - | - |
| MFLVC | - | - | - | - | - | - | - |
| UOMVSC | 66.26($\pm$0.00) | 38.84($\pm$0.00) | 30.45($\pm$0.00) | 19.29($\pm$0.00) | 26.28($\pm$0.00) | - | - |
| PGSC | 64.94($\pm$7.61) | 36.37($\pm$4.52) | 31.27($\pm$2.43) | 18.07($\pm$0.74) | 22.15($\pm$0.62) | - | - |
| DSMVC | - | - | - | - | - | - | - |
| AEVC | - | - | - | - | - | - | - |
| OrthNTF | - | - | - | - | - | - | - |
| FMVACC | - | - | - | - | - | - | - |
| FASTMI | 64.58($\pm$4.53) | 37.46($\pm$1.93) | 34.52($\pm$1.21) | **21.73($\pm$0.91)** | 21.79($\pm$2.80) | 23.21($\pm$1.05) | 53.12($\pm$4.21) |
| Ours | 66.76($\pm$0.00) | **52.04($\pm$0.00)** | **35.78($\pm$0.00)** | 20.06($\pm$0.00) | **28.03($\pm$0.00)** | **25.73($\pm$0.00)** | **56.92($\pm$0.00)** |

## M. The Effectiveness of Single View Scenarios

Except for multi-view scenarios, sometimes we also may encounter the datasets containing only one view. To validate the ability to tackle single view scenarios, we conduct clustering operation on one view rather than on all views of datasets mentioned earlier. The experimental results are summarized in Table 13, 14 and 15. From these tables, we can draw

*Table 14.* Single-view experimental results in NMI

| Dataset | DERMATO | CALTE7 | Cora | REU7200 | Reuters | CIF10Tra4 | FasMNI4V |
|---|---|---|---|---|---|---|---|
| FMR | 76.63(±3.98) | 1.38(±0.18) | 5.12(±0.17) | - | - | - | - |
| PMSC | 80.21(±5.10) | 34.04(±0.71) | 6.12(±0.83) | 1.43(±0.12) | - | - | - |
| AMGL | 3.07(±0.24) | 34.34(±1.28) | 0.83(±0.02) | 0.72(±0.09) | 0.89(±0.03) | - | - |
| MSCIAS | 75.79(±4.77) | 35.42(±1.66) | 9.45(±0.46) | 1.69(±0.27) | 1.22(±0.17) | - | - |
| MVSC | 52.19(±11.98) | 25.22(±1.25) | - | - | - | - | - |
| MLRSSC | 0.54(±0.00) | 0.73(±0.00) | 0.48(±0.00) | 0.56(±0.00) | - | - | - |
| MPAC | 77.69(±0.00) | 29.88(±0.00) | 9.30(±0.00) | 1.29(±0.00) | - | - | - |
| MCLES | 78.19(±3.49) | 27.64(±1.87) | 8.47(±0.93) | - | - | - | - |
| FMCNOF | 42.62(±4.32) | 9.42(±1.23) | 4.58(±0.79) | 1.11(±0.17) | - | 9.36(±2.32) | 32.07(±3.26) |
| ADAGAE | - | - | - | - | - | - | - |
| PFSC | 76.00(±2.45) | 32.47(±1.85) | - | - | - | - | - |
| SFMC | 62.77(±0.00) | 27.46(±0.00) | 6.01(±0.00) | 2.37(±0.00) | 1.44(±0.00) | 9.44(±0.00) | - |
| MSGL | 10.33(±0.74) | - | - | 1.15(±0.08) | 1.02(±0.06) | 7.64(±0.68) | - |
| DEMVC | - | - | - | - | - | - | - |
| FPMVS | 77.14(±5.56) | 34.02(±1.85) | 9.28(±1.64) | 2.44(±0.24) | **11.73(±3.38)** | 10.99(±0.82) | 53.71(±2.18) |
| 3AMVC | - | - | - | - | - | - | - |
| MFLVC | - | - | - | - | - | - | - |
| UOMVSC | 77.83(±0.00) | 28.86(±0.00) | 8.95(±0.00) | 2.83(±0.00) | 9.45(±0.00) | - | - |
| PGSC | 69.96(±4.66) | 22.46(±2.37) | 1.43(±0.32) | 2.03(±0.26) | 0.92(±0.08) | - | - |
| DSMVC | - | - | - | - | - | - | - |
| AEVC | - | - | - | - | - | - | - |
| OrthNTF | - | - | - | - | - | - | - |
| FMVACC | - | - | - | - | - | - | - |
| FASTMI | 78.85(±2.10) | 32.49(±1.29) | 9.77(±2.13) | **3.03(±0.66)** | 10.12(±1.58) | 13.48(±0.64) | 58.14(±1.73) |
| Ours | **80.52(±0.00)** | **36.59(±0.00)** | **10.04(±0.00)** | 2.50(±0.00) | 1.37(±0.00) | **15.09(±0.00)** | **61.44(±0.00)** |

*Table 15.* Single-view experimental results in Fscore

| Dataset | DERMATO | CALTE7 | Cora | REU7200 | Reuters | CIF10Tra4 | FasMNI4V |
|---|---|---|---|---|---|---|---|
| FMR | 66.40(±4.64) | 22.72(±0.15) | 19.09(±0.23) | - | - | - | - |
| PMSC | 68.06(±8.46) | 37.68(±1.18) | 27.42(±2.21) | 21.34(±1.02) | - | - | - |
| AMGL | 19.17(±0.28) | 37.52(±0.89) | 24.77(±0.11) | 22.53(±0.00) | 28.53(±0.21) | - | - |
| MSCIAS | 67.19(±8.74) | 40.65(±3.12) | 23.41(±1.52) | 21.07(±0.41) | 30.72(±0.77) | - | - |
| MVSC | 55.63(±10.99) | 44.64(±2.87) | - | - | - | - | - |
| MLRSSC | 33.39(±0.00) | **50.64(±0.00)** | 30.40(±0.00) | 23.21(±0.01) | - | - | - |
| MPAC | **70.02(±0.00)** | 41.65(±0.00) | 26.10(±0.00) | 21.94(±0.00) | - | - | - |
| MCLES | 67.27(±4.69) | 48.97(±2.84) | 30.34(±1.73) | - | - | - | - |
| FMCNOF | 46.98(±3.78) | 46.35(±4.21) | 20.05(±2.23) | 21.68(±2.79) | - | 17.00(±1.17) | 26.54(±1.89) |
| ADAGAE | - | - | - | - | - | - | - |
| PFSC | 68.99(±4.95) | 48.47(±3.26) | - | - | - | - | - |
| SFMC | 60.29(±0.00) | 47.32(±0.00) | 30.35(±0.00) | 20.64(±0.00) | 29.04(±0.00) | 15.28(±0.00) | - |
| MSGL | 31.90(±0.97) | - | - | 22.48(±0.37) | 28.97(±0.24) | 16.14(±0.32) | - |
| DEMVC | - | - | - | - | - | - | - |
| FPMVS | 68.31(±6.97) | 49.97(±2.50) | 26.33(±0.70) | 20.33(±1.15) | 31.80(±1.64) | 17.54(±0.35) | 45.93(±2.06) |
| 3AMVC | - | - | - | - | - | - | - |
| MFLVC | - | - | - | - | - | - | - |
| UOMVSC | 67.90(±0.00) | 38.64(±0.00) | 24.19(±0.00) | 22.78(±0.00) | 31.91(±0.00) | - | - |
| PGSC | 63.05(±6.85) | 37.62(±4.23) | 25.01(±3.77) | 22.53(±1.24) | 32.42(±1.53) | - | - |
| DSMVC | - | - | - | - | - | - | - |
| AEVC | - | - | - | - | - | - | - |
| OrthNTF | - | - | - | - | - | - | - |
| FMVACC | - | - | - | - | - | - | - |
| FASTMI | 69.65(±4.79) | 39.57(±1.24) | 31.67(±1.50) | 19.24(±1.49) | 30.45(±0.71) | 15.35(±0.69) | 45.68(±2.35) |
| Ours | 69.70(±0.00) | 50.41(±0.00) | **32.40(±0.00)** | **25.26(±0.00)** | **35.02(±0.00)** | **17.86(±0.00)** | **49.77(±0.00)** |

that ADAGE, DEMVC, 3AMVC, MELVC, DSMVC, AEVC, OrthNTF and FMVACC are powerless against single view scenarios, which is mainly because they generally need utilize the information of other views to help optimize. FMR, PMSC,

AMGL, MSCIAS, MVSC, etc, are able to work properly with single view scenarios, nevertheless, they generally produce inferior clustering results in most situations. By comparison, besides being able to operate properly on single view scenarios, our DTP-SF-BVF also can generate desirable results. Accordingly, our DTP-SF-BVF enjoys wider serviceability.

*Table 16.* The effectiveness of view information gathering

| Dataset | Metric | Clustering Results | | | | | | |
|---|---|---|---|---|---|---|---|---|
| | | V1 | V2 | V3 | V4 | V5 | V6 | Ours |
| DERMATO | ACC | 60.06 | 66.76 | | | | | **85.47** |
| | NMI | 56.66 | 80.52 | | | | | **89.97** |
| | Fscore | 49.99 | 69.70 | | | | | **87.92** |
| CALTE7 | ACC | 35.01 | 52.04 | 48.71 | 53.87 | 39.42 | 46.40 | **80.66** |
| | NMI | 17.84 | 36.59 | 34.14 | **48.75** | 41.74 | 39.98 | 45.25 |
| | Fscore | 33.35 | 50.41 | 47.41 | 53.85 | 48.15 | 47.42 | **78.12** |
| Cora | ACC | 30.24 | 35.78 | 30.17 | 29.73 | | | **52.44** |
| | NMI | 10.22 | 10.04 | 12.24 | 11.49 | | | **43.70** |
| | Fscore | 30.39 | 32.40 | 30.36 | 29.91 | | | **41.12** |
| REU7200 | ACC | 21.24 | 20.06 | 24.06 | 19.61 | 20.83 | | **26.22** |
| | NMI | 1.27 | 2.50 | 4.81 | 1.39 | 1.40 | | **6.25** |
| | Fscore | 23.67 | 25.26 | 25.41 | 27.01 | 24.90 | | **28.55** |
| Reuters | ACC | 46.65 | 28.03 | 27.24 | 27.80 | 47.62 | | **54.26** |
| | NMI | 23.47 | 1.37 | 1.10 | 1.06 | 20.83 | | **31.87** |
| | Fscore | 42.71 | 35.02 | 35.24 | 35.16 | 43.67 | | **44.84** |
| CIF10Tra4 | ACC | 21.89 | 25.73 | 23.25 | 22.05 | | | **26.83** |
| | NMI | 10.01 | 15.09 | 12.35 | 12.50 | | | **15.64** |
| | Fscore | 16.46 | 17.86 | 15.87 | 16.34 | | | **20.64** |
| FasMNI4V | ACC | 41.76 | 56.92 | 46.71 | 52.32 | | | **57.36** |
| | NMI | 49.81 | **61.44** | 53.26 | 54.86 | | | 59.21 |
| | Fscore | 37.26 | 49.77 | 42.80 | 45.04 | | | **51.37** |

## N. The Effectiveness of View Information Gathering

Compared to single view datasets, multi-view data can provide more comprehensive and detailed descriptions for the same instance and thereby facilitates more accurate representations for better clustering. To validate the effectiveness of DTP-SF-BVF in gathering the information from multiple views, on the basic of Section M, we conduct clustering individually on each view of multi-view datasets mentioned earlier and compare the generated single-view clustering results and multi-view clustering results, as shown in Table 16, where V1 ∼ V6 denote the results based on view 1 ∼ 6 respectively and 'Ours' denotes the results based on all views. As seen, multi-view clustering results outperform single-view counterparts with remarkable margins in most cases, which highlights that our DTP-SF-BVF is able to effectively gather multi-view information for preferable clustering. The reason of some sub-optimal results could be that the quality of certain views is relatively poor and disorganize the cluster structure.

## O. Time Overhead Proportion

To further dissect the performance of the proposed DTP-SF-BVF, we count the time overhead proportion of each optimization variable, as shown in Fig. 4. From these figures, we can observe that on CALTE7 and Cora datasets, $\mathbf{B}_p$ and $\mathbf{T}_p$ occupy most of the overall optimization time, which is mainly because the number of clusters is slightly larger and accordingly the traversal searching and QCQP searching consume relatively more time than other parts. On REU7200 and Reuters, the time overhead of $\mathbf{C}$ and $\alpha$ holds a dominant position, possibly because the higher data dimension exacerbates the computing

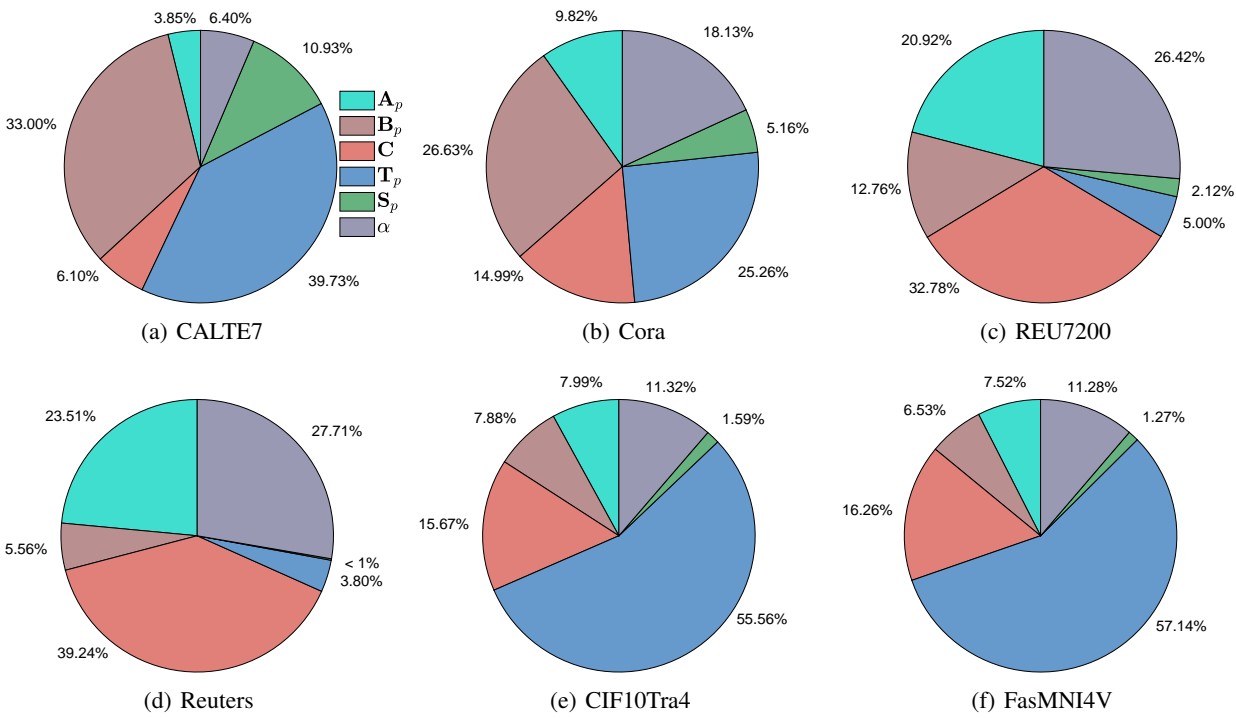

*Figure 4.* The time consumption ratio of different optimization variables.

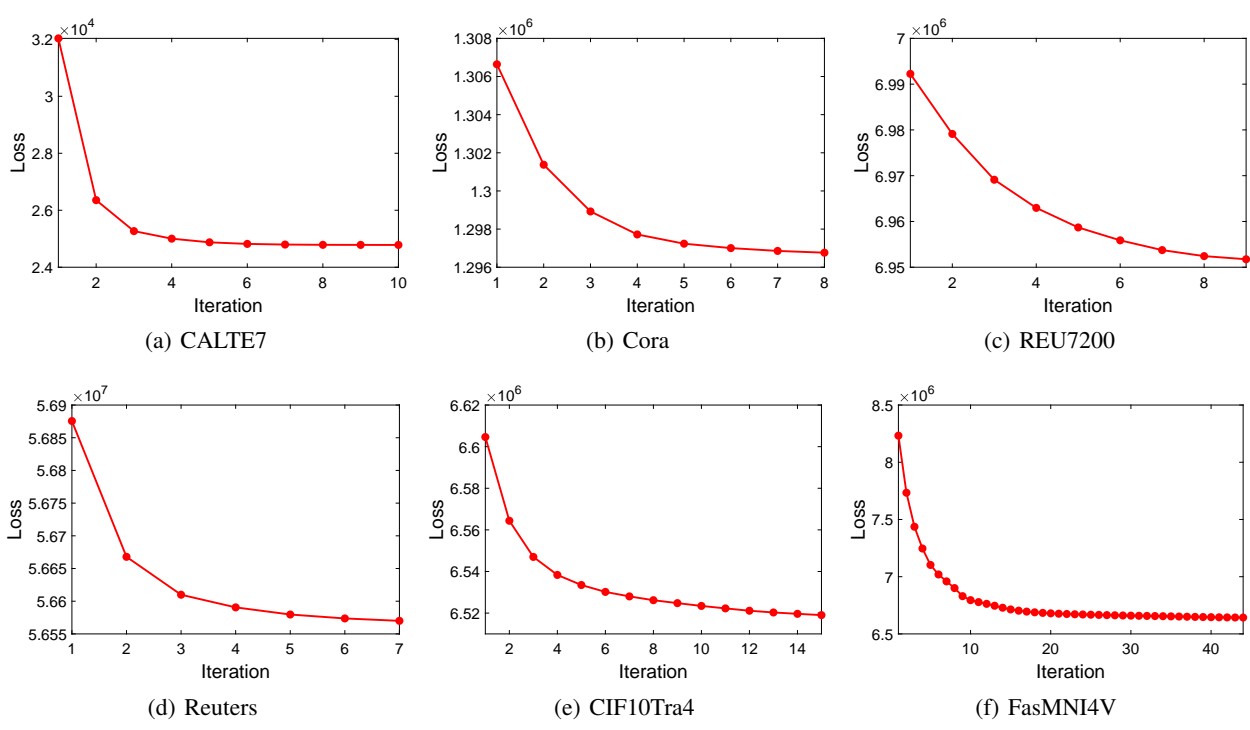

*Figure 5.* Changes in the loss function.

burden of $\mathbf{W}$, $\mathbf{Z}$ and the coefficient $b_p$. When dealing with CIF10Tra4 and FasMNI4V, the time overhead of updating $\mathbf{T}_p$ and $\mathbf{C}$ is larger than that of other variables. Possible reasons are that the cluster number and the feature dimension on

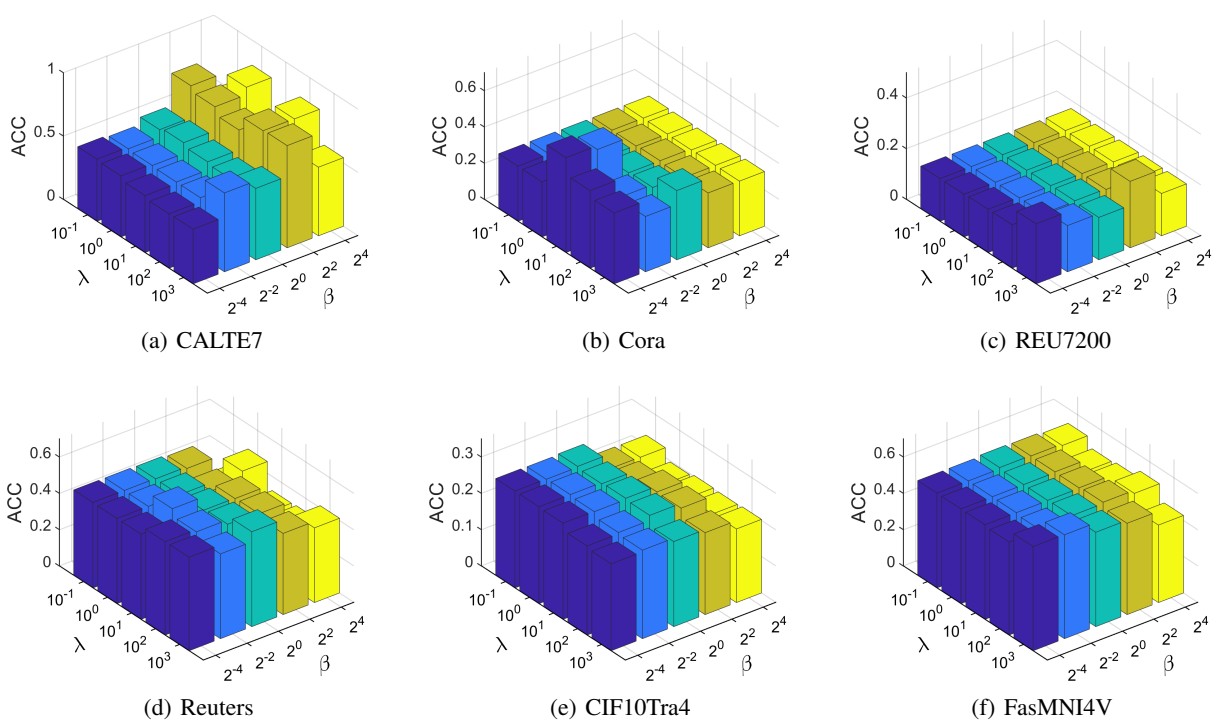

*Figure 6.* Sensitivity of the parameters $\lambda$ and $\beta$ in terms of ACC.

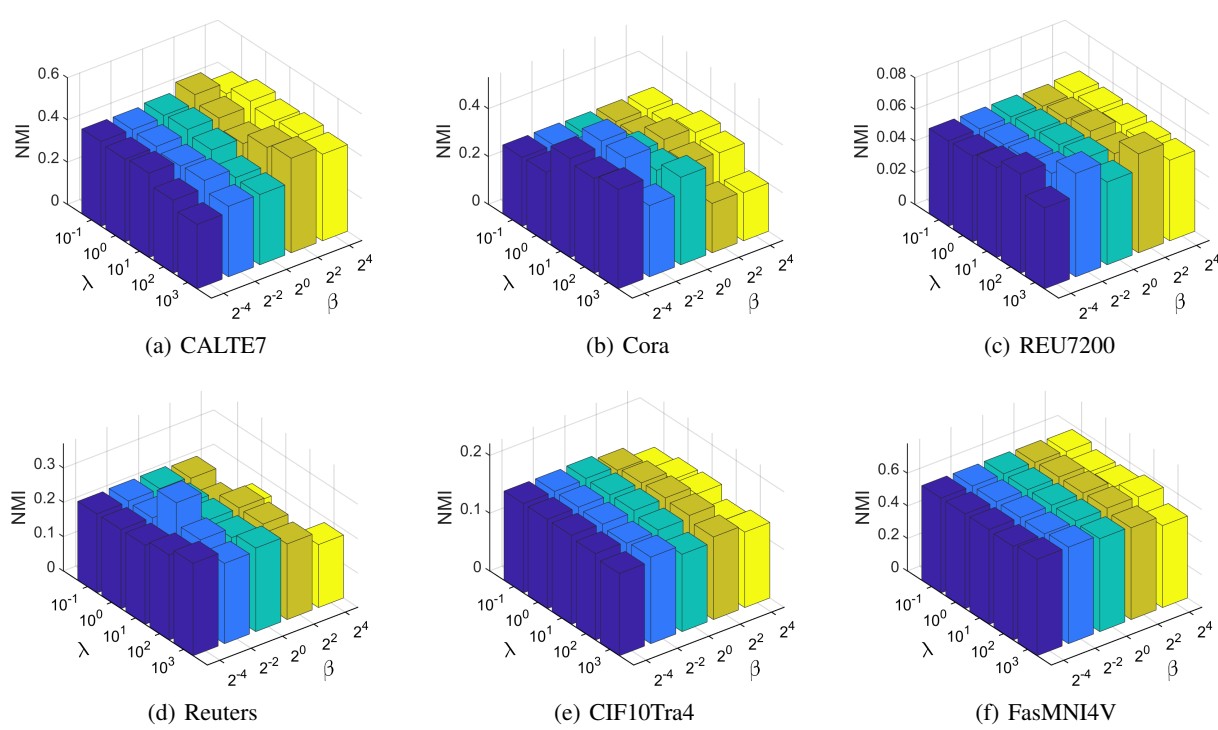

*Figure 7.* Sensitivity of the parameters $\lambda$ and $\beta$ in terms of NMI.

these two datasets are relatively larger and accordingly induces much time overhead. Especially, $\mathbf{T}_p$ takes the most time expenditures, which is mainly due to the searching on a set of one-hot vectors. Although the time consumption proportion

between optimization variables is diverse in different cases, combined with Fig. 2 we have that the overall time consumption of our DTP-SF-BVF is competitive.

## P. Convergence

In addition to owing linear complexities, our DTP-SF-BVF is also convergent. To demonstrate this point, we plot the changes in loss function with respective to the number of iterations, as shown in Fig. 5. One can observe that the function loss is monotonically reducing after iterations and gradually reaches to a steady state, which gives evidence that the proposed DTP-SF-BVF is convergent.

## Q. Sensitivity

In our DTP-SF-BVF method, there involve hyper-parameters $\lambda$ and $\beta$. We conduct fine tuning for them in $[10^{-1}, 10^0, \cdots, 10^3]$ and $[2^{-4}, 2^{-2}, \cdots, 2^4]$ respectively. To investigate the sensitivity of hyper-parameters $\lambda$ and $\beta$, we plot the clustering results under each parameter combination, as shown in Fig. 6, 7 and 8. It is easy to see that with given $\beta$, the clustering results are not dramatically changed in most cases. So, we can conclude that the proposed DTP-SF-BVF is not fairly sensitive to $\lambda$. Moreover, combined with Table 1, we have that within a broad range of parameters, the generated clustering results are still comparable. Thus, we can summarize that the proposed DTP-SF-BVF is somewhat robust to hyper-parameters.

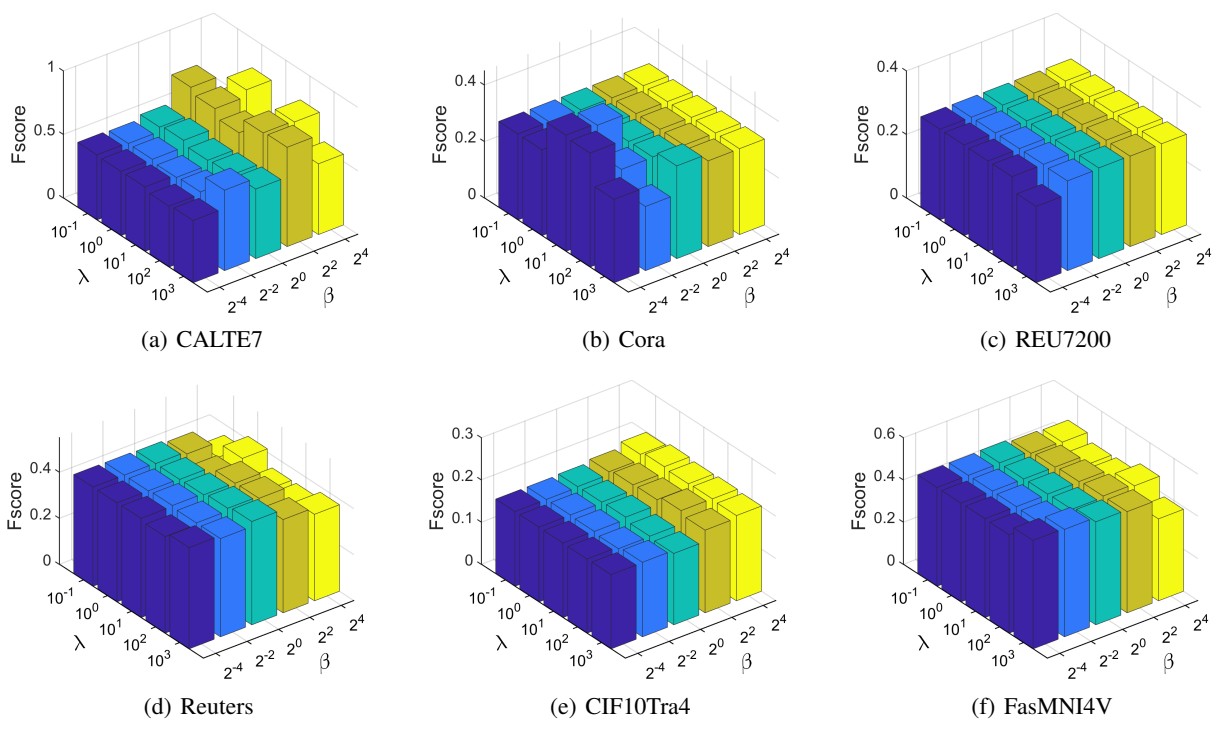

*Figure 8.* Sensitivity of the parameters $\lambda$ and $\beta$ in terms of Fscore.

## R. Potential Improvement Directions

In this work, we generate anchors via learning strategy, nevertheless, we do not explicitly consider the spatial distribution of anchors. Given the fact that the role of anchors aims at approximately characterizing the overall samples, generating the anchors that are with similar distributions to original data could further enhance the clustering performance. Besides, it needs to perform searching on one-hot vectors when updating the permutation model, which could bring additional computing overhead, and thus designing other talented solutions will further accelerate its running speed.

In addition to these, the tensor Schatten $p$-norm is usually regarded as a good means to exploit the complementary information

between views. Our model at present dose not involve the Schatten $p$-norm, and adopts the shared cluster indicator matrix to capture view complementary information. Including the Schatten $p$-norm could help further enhance the clustering ability of our model in the future.

