# OpenReview forum: "From Spectrum-free towards Baseline-view-free: Double-track Proximity Driven Multi-view Clustering"
_ICML.cc/2025/Conference — ICML 2025 poster_

### Official Review · Reviewer_Mxef · 2025-03-12

**Overall Recommendation:** 4

**Summary:**

This work proposes a spectrum-free and baseline-view-free multi-view clustering method with double-track proximity, DTP-SF-BVF.  It aims at improving the clustering stability and alignment flexibility as well as the anchor itself characteristic exploration. Unlike current methods that usually overlook the proximity relationship between anchors, this work utilizes self-expression learning and point-point topology learning to capture that. To get rid of the limitation in baseline view, this work designs a learnable permutation strategy to jointly reorder anchors according to respective view characteristics. To alleviate the variance impact, this work avoids the formulating of spectrum. Experiments on multiple benchmark datasets demonstrate the effectiveness of the presented DTP-SF-BVF.

**Claims And Evidence:**

The claims are clear.

**Essential References Not Discussed:**

None

**Experimental Designs Or Analyses:**

I have checked the soundness of experimental designs.

**Methods And Evaluation Criteria:**

The proposed DTP-SF-BVF method can effectively handle the multi-view clustering problem and be applied to large-scale scenes.

**Other Comments Or Suggestions:**

None

**Other Strengths And Weaknesses:**

Its strengths are as follows,

-  The writing is easy to follow, and the organization is clear.

-  The solving procedure presented in the paper is detailed.

-  Experiments conducted are sufficient and reveal the effectiveness of proposed method from multiple perspectives.


Its weaknesses are as follows,

-   Clearly outlining the final steps for deriving clustering results and specifying the initialization settings of relevant variables are crucial for ensuring a thorough understanding of the algorithm's workflow.

-   Offering theoretical explanations for the sub-optimal results generated would significantly deepen the understanding of the algorithm's underlying mechanisms and limitations.

**Questions For Authors:**

-   The current methodology involves constructing individual anchor graphs for each view, while why not directly learn a unified anchor graph matrix? After aligning, could this potentially provide a more superior clustering accuracy?

-   While the experimental results in the paper are impressive, the interpretability of the clustering outputs and the practical significance could benefit from additional exploration. The authors could provide valuable insights by discussing the interpretation of clustering results in real-world applications, highlighting the method's potential for broader impact.

-  When clustering different kinds of datasets, which factors are more prone to restrict the proposed algorithm?  dataset size, feature dimension, or hyper-parameter?

-  While this approach demonstrates potential advantages, it is noteworthy that several recent MVC techniques have effectively circumvented the need for alignment by leveraging shared features. Whether are there particular scenarios where the necessity for such alignment could be reduced? The authors are encouraged to provide a more detailed exposition on the alignment criticality.

**Relation To Broader Scientific Literature:**

The work proposes an interesting multi-view clustering, which is with spectrum-free and baseline-view-free as well as double-track proximity properties.  Besides these, it is with linear time complexity and linear space complexity.

**Theoretical Claims:**

I have checked the derivation procedure.

---

> ### Author Rebuttal · Authors · 2025-03-31
>
> **Q1:** Final steps for deriving clustering and variable initialization.
>
> **A1:** After getting $\mathbf{C}$, we derive the clustering by sequentially identifying the row numbers where the element 1 is located. We initialize $\mathbf{A}_p$, $\mathbf{T}_p$, $\mathbf{B}_p$ and $\boldsymbol{\alpha}$ with random matrix, unit matrix, orthogonal matrix and $1/v$. For $\mathbf{C}$, we create a zero matrix and then randomly assign a single 1 to each column. For $\mathbf{S}_p$, we assign elements ranging from 0 to 1 column by column and ensure that the diagonal elements remain 0 and the sum of each column equals 1.
>
> **Q2:** Explanations for sub-optimal results.
>
> **A2:** Thanks! MSCIAS achieves a marginally superior Fscore (0.72\%) on Cora, possibly because of the introduction of HSIC and the employment of local connectivity. FPMVS achieves a 0.29\% accuracy gain on CIF10Tra4 owing to the adoption of orthogonal projection ensembles. SFMC integrates a structural coherence and self-supervised attention. MFLVC introduces hierarchical feature learning and consensus semantics.
>
> **Q3:** Why not learn a unified anchor graph? Performance.
>
> **A3:** Anchor graphs on respective views may be more conducive to expressing the characteristics of their own views. A single graph structure may not adequately capture the intrinsic characteristics across all views. To confirm this, we conduct relevant experiments. IAG and UAG are the results based on individual and unified anchor graph respectively.
>
> |Dataset|DERMATO|CALTE7|Cora|REU7200|Reuters|CIF10Tra4|FasMNI4V|
> |:---:|:---:|:---:|:---:|:---:|:---:|:---:|:---:|
> |ACC||||||||
> |UAG|83.58|74.38|49.43|25.44|51.81|23.27|53.31|
> |IAG|**85.47**|**80.66**|**52.44**|**26.22**|**54.26**|**26.83**|**57.36**|
> |NMI||||||||
> |UAG|85.37|43.92|42.07|6.17|29.74|13.93|54.37|
> |IAG|**89.97**|**45.25**|**43.70**|**6.25**|**31.87**|**15.64**|**59.21**|
> |Fscore||||||||
> |UAG|83.72|70.43|**42.63**|25.79|**46.83**|19.18|**52.93**|
> |IAG|**87.92**|**78.12**|41.12|**28.55**|44.84|**20.64**|51.37|
>
> **Q4:** Interpretation about the clustering outputs and the practical significance.
>
> **A4:** Our model introduces a alignment mechanism without the necessity of selecting a baseline view. It is adept at synchronizing with the anchor generation, and adeptly rearranges anchors within their original space, well preserving the data diversity. Our model  incorporates the geometric properties among anchors into the anchor-sample similarity, more thoroughly uncovering the manifold structure inherent into samples. Besides, our model engages in the direct learning of consensus cluster indicators, consolidating  multi-view information at the cluster-label level. Further, our model is equipped with linear complexity. Owing to these, our model yields stable results and is adept at handling large-scale scenes.
>
> **Q5:**  Restriction factors for the proposed method.
>
> **A5:** The computing cost is $\mathcal{O}(m^2nv+dnm+m!v+m^3kv)$. In general, $m$, $v$ and $k$ are greatly smaller than $n$. $d$ is a constant and irrelevant to $n$. The computing cost is linear with respect to $n$. Compared to the square and cube, the factorial about $m$ needs more cost. Too large $m$ will induce expensive cost. In addition, the space overhead is $\mathcal{O}(nk)$. So, the proposed method is not limited by its space overhead. Further, combined with the sensitivity study, the performance is relatively robust to hyper-parameters. So, the method is mainly influenced by the anchor number.
>
> **Q6:** Exposition on the alignment criticality.
>
> **A6:** The works based on shared features (SF) typically derive  consensus anchors rather than view-tailored  anchors to establish similarity. This harnesses the complementary. The following  experiments further show this.
>
> |||||||||
> |:---:|:---:|:---:|:---:|:---:|:---:|:---:|:---:|
> |ACC||||||||
> |SF|81.23|71.33|49.73|24.97|48.46|22.36|53.38|
> |NMI||||||||
> |SF|82.76|42.26|39.87|6.03|29.89|13.43|51.97|
> |Fscore||||||||
> |SF|80.64|69.72|**42.28**|25.21|**46.13**|19.58|**52.21**|
>
> In our model, the alignment mechanism builds pure self-expression affinities. If not aligning, the structure of  anchor-anchor affinity would be chaotic, which will in turn impair  the anchor-sample proximity, hindering the clustering performance. The following experiments validate this point. WOA is the results without our alignment.
>
> |||||||||
> |:---:|:---:|:---:|:---:|:---:|:---:|:---:|:---:|
> |ACC||||||||
> |WOA|80.73|76.59|31.65|16.67|45.29|25.91|53.68|
> |NMI||||||||
> |WOA|82.53|39.55|35.41|3.32|24.77|15.30|56.47|
> |Fscore||||||||
> |WOA|79.47|72.23|30.69|21.14|42.59|17.90|47.41|
>
> In cases where the consistent information predominates over the complementary, it may be feasible to establish unified anchors to formulate similarity and thus potentially circumvent or mitigate the need for alignment. Nonetheless, given the inherent complexity of multi-view data, accurately quantifying the complementary and consistent information is usually a challenging task.

---

> > ### Comment · Reviewer_Mxef · 2025-04-07
> >
> > The author's response has addressed my concerns. I have therefore decided to raise my score.

---

> > > ### Author Response · Authors · 2025-04-08
> > >
> > > Thanks!!

---

### Official Review · Reviewer_Les7 · 2025-03-12

**Overall Recommendation:** 3

**Summary:**

In this paper, the authors concentrate on three key issues in multi-view clustering field:  the neglect of anchor-anchor geometric proximity, the reliance on the baseline view for anchor alignment, and the instability caused by spectrum. Firstly, the authors adopt a self-expression subspace skill to explicitly exploit anchor characteristics and feed them into similarity graph via topology learning to explore manifold structure inside samples. Then, they introduce a joint permutation mechanism, eliminating the requiring of baseline view and concurrently working with the generation of anchors. Furthermore, they design a consensus discrete structure, which skips the spectrum, to directly produce cluster labels and meanwhile provide a common link to facilitate anchor transformation.

**Claims And Evidence:**

Yes. The claims in this paper are verified by experiments and discussions.

**Essential References Not Discussed:**

No.

**Experimental Designs Or Analyses:**

Yes. The experimental designs follows the commonly-used settings and the results are also reliable.

**Methods And Evaluation Criteria:**

Yes, It makes sense.

**Other Comments Or Suggestions:**

See the following questions.

**Other Strengths And Weaknesses:**

Strengths:

(1) The analysis about experimental results is in-depth.

(2) The linear time and space complexities of designed model guarantee its practicality.

(3) The review about existing literatures is comprehensive.


Weaknesses:

(1) During optimizing the permutation variable on each view, the introduction about its computational cost seems a bit concise, specially, the searching on one-hot vectors.

(2) In Algorithm 1, the stopping condition is missing.  It would be helpful for authors to explicitly describe it.

(3) To improve the accessibility and visual appeal of the dataset introduction, employing a table format would be highly advantageous.

**Questions For Authors:**

(1) Rather than employing view-related anchors to extract data features, when adopting consensus anchors, the misalignment could not exist since all views share a common group of anchors. Under this situation, how is the performance? Is the permutation necessary?

(2) From Eq.(8) to Eq.(9), why does this transformation hold?

(3) How to determine/set the value of $m$? The parameter $m$ may affect the clustering performance since it leads to graphs with diverse scales.

(4)  According to the ablation results about the spectrum-free strategy, one can obtain that it takes less time cost than CS. why? what are possible reasons for this phenomenon?

**Relation To Broader Scientific Literature:**

This paper integrates the proximity between anchors into anchor graph to more adequately   characterize the manifold structure inside samples, and designs a spectrum-free multi-view clustering paradigm without requiring baseline-view.

**Theoretical Claims:**

The Appendix provides the update rules and derivations for variables.

---

> ### Author Rebuttal · Authors · 2025-03-31
>
> **Q1:** The introduction about the computational cost of permutation seems a bit concise.
>
> **A1:** More details are provided here.  Due to $\mathbf{A}_p\in{d_p\times m}$, $\mathbf{S}_p\in{m\times m}$, $\mathbf{B}_p\in{m\times k}$, $\mathbf{C}\in{k\times n}$ and $\mathbf{X}_p\in{d_p\times n}$, building $\mathbf{G}_p$, $\mathbf{H}_p$, $\mathbf{M}_p$ and $\mathbf{J}_p$ will require $\mathcal{O}(d_pm^2)$, $\mathcal{O}(m^3)$, $\mathcal{O}(mkn+m^2n)$ and $\mathcal{O}(d_pmn+mnk+m^2k)$ cost. Due to $\mathbf{T}_p\in{m\times m}$ consisting of 0 and 1, conducting traversal searching on one-hot vectors will take $\mathcal{O}(m!)$.  So, updating the permutation takes $\mathcal{O}(d_pm^2+d_pmn+m^3+mkn+m^2n+m!)$.
>
> **Q2:** In Algo 1, the stopping condition is missing.
>
> **A2:** Thanks! We run the Algo 1 under $f(t)-f(t+1)<=10^{-3}f(t)$. $f(t)$ is the objective value at $t$-th iteration.
>
> **Q3:** Employing a table format to introduce datasets.
>
> **A3:** Good advice! We will present that in a table format to enhance its visual appeal.
>
> **Q4:** Under consensus anchors, how is the performance? Is the permutation necessary?
>
> **A4:** The paradigm based on consensus anchors extract a set of common anchors rather than multiple sets of view-specific anchors to build similarity. Although avoiding alignment, due to the lack of view-unique characteristics, this could not extract sufficient diverse features. To further demonstrate this, we organize  experiments. CA and VA are the results based on consensus and view-related anchors respectively.
>
> |Dataset|DERMATO|CALTE7|Cora|REU7200|Reuters|CIF10Tra4|FasMNI4V|
> |:---:|:---:|:---:|:---:|:---:|:---:|:---:|:---:|
> |ACC||||||||
> |CA|81.23|71.33|49.73|24.97|48.46|22.36|53.38|
> |VA|**85.47**|**80.66**|**52.44**|**26.22**|**54.26**|**26.83**|**57.36**|
> |NMI||||||||
> |CA|82.76|42.26|39.87|6.03|29.89|13.43|51.97|
> |VA|**89.97**|**45.25**|**43.70**|**6.25**|**31.87**|**15.64**|**59.21**|
> |Fscore||||||||
> |CA|80.64|69.72|**42.28**|25.21|**46.13**|19.58|**52.21**|
> |VA|**87.92**|**78.12**|41.12|**28.55**|44.84|**20.64**|51.37|
>
> The reason of this could be that the view-exclusive representation that view-related (aligned) anchors contain outweighs the view-common information that consensus anchors contain.
>
> **Q5:** From Eq.(8) to Eq.(9), why does it hold?
>
> **A5:** For the objective, we have
> $\operatorname{Tr} \left(\mathbf{B} _p^{\top}\mathbf{H}_p\mathbf{B} _p \mathbf{C}\mathbf{C}^{\top} \right)  \Leftrightarrow [\mathbf{B} _p^{\top}] _{j,:} [\mathbf{H}_p\mathbf{B} _p \mathbf{C}\mathbf{C}^{\top}] _{:,j}  \Leftrightarrow  [\mathbf{B} _p^{\top}] _{j,:}\mathbf{H}_p\mathbf{B} _p[\mathbf{C} \mathbf{C}^{\top}] _{:,j}$ and $\operatorname{Tr}\left(\boldsymbol{\alpha} _p^2\mathbf{C}\mathbf{X} _p^{\top}\mathbf{A} _p \mathbf{T} _p \mathbf{B} _p\right) \Leftrightarrow \left[\boldsymbol{\alpha} _p^2\mathbf{C}\mathbf{X} _p^{\top}\mathbf{A} _p\mathbf{T} _p\right] _{j,:}[\mathbf{B} _p] _{:,j}$ where we omit the $\min$ operator and $\mathbf{H}_p=\beta\mathbf{L _s}+\boldsymbol{\alpha} _p^2\mathbf{Q} _p$. $\mathbf{C}\mathbf{C}^{\top}$ is diagonal, and we have $[\mathbf{B}_p^{\top}] _{j,:}\mathbf{H}_p\mathbf{B} _p[\mathbf{C}\mathbf{C}^{\top}] _{:,j} \Leftrightarrow [\mathbf{B} _p] _{:,j}^{\top}\sum _{i=1}^n \mathbf{C} _{j,i}\mathbf{H}_p[\mathbf{B} _p] _{:,j}$.
> For the feasible region, $\mathbf{B} _p^{\top}\mathbf{B}_p=\mathbf{I} _k$ can be divided into
> $[\mathbf{B} _p] _{:,j}^{\top}[\mathbf{B} _p] _{:,j}=1$ and $[\mathbf{B} _p] _{:,j}^{\top}[\mathbf{B} _p] _{:,i}=0,i=1,2, \cdots,k,i\neq j,j=1,2,\cdots,k$. Further, $[\mathbf{B} _p] _{:,j}^{\top}[\mathbf{B} _p] _{:,j}=1$ can be written as $[\mathbf{B} _p] _{:,j}^{\top}\mathbf{I} _{m\times m}[\mathbf{B} _p] _{:,j}-1=0$. The $[\mathbf{B} _p] _{:,j}^{\top}[\mathbf{B} _p] _{:,i}=0,i=1,2,\cdots,k,i\neq j$ can be written as $\left[[\mathbf{B} _p] _{:,1},[\mathbf{B} _p] _{:,2},\cdots,[\mathbf{B} _p] _{:,j-1},[\mathbf{B} _p] _{:,j+1},\cdots,[\mathbf{B} _p] _{:,k}\right]^{\top}[\mathbf{B} _p] _{:,j}=\mathbf{0} _{(k-1)\times 1}$.
>
> **Q6:** How to set $m$?
>
> **A6:** We set it equal to the number of clusters.
> During updating $\mathbf{T}_p$, the objective form is $\operatorname{Tr}\left(\mathbf{T} _p^\top\mathbf{B}\mathbf{T} _p\mathbf{C}+\mathbf{T} _p^{\top}\mathbf{D}\right)$. Besides, the feasible region is discrete. These  cause it being hard to solve. To this end, we adopt the traversal searching on one-hot vectors to obtain the optimal solution. This takes $\mathcal{O}(m!)$ computing cost. Too large $m$ will induce intensive time.
>
> **Q7:** Why taking less time than CS? Possible reasons?
>
> **A7:** CS needs to first form spectrum and then conduct embedding partitioning. These two procedures induce extra computing cost. On the other hand, we directly generate labels via binary learning. The labels can be get in a closed-form solution. It only compares the diagonal elements of $\mathbf{W}$ and the row elements of $\mathbf{Z}$. The element scale is $k$, which is small-sized.

---

### Official Review · Reviewer_5v1e · 2025-03-13

**Overall Recommendation:** 3

**Summary:**

This paper develops a multi-view clustering algorithm named DTP-SF-BVF to address the problems: (1) current methods usually focus only on the anchor-sample proximity and fail to take into account the anchor-anchor relationship; (2) they require to select the baseline view; (3) existing spectrum paradigm induces clustering variance. DTP-SF-BVF leverages double-track proximity to extract both anchor-anchor and anchor-sample characteristics hidden into data, and devises a permutation mechanism for each view, eliminating the need for selecting the baseline view. Further, it chooses to directly formulate the label indicators through a consensus structure, and in turn the consensus structure bridges anchors on all views.  Extensive experiments validate DTP-SF-BVF’s effectiveness.

**Claims And Evidence:**

Yes, the motivation of double-track proximity, baseline-view-free, and spectrum-free paradigm is clearly claimed.

**Essential References Not Discussed:**

The references are discussed sufficiently.

**Experimental Designs Or Analyses:**

I checked the experimental results and discussions.

**Methods And Evaluation Criteria:**

Yes, it is suitable for multi-view clustering tasks.

**Other Comments Or Suggestions:**

No other comments or suggestions.

**Other Strengths And Weaknesses:**

**Strengths:**
1. The idea is well-motivated.  The double-track proximity paradigm effectively enhances the clustering accuracy via utilizing a more comprehensive view of data representation.
2. Baseline-view-free and spectrum-free schemes highlight the flexibility and reliability of DTP-SF-BVF for clustering data from diverse sources.
3. The organized experiments are sufficient.

**Weaknesses:**
1.  The binary elements inherent in the alignment model introduce complexities. Consequently, the question arises as to whether this model can be further refined. Specially, the orthogonality possesses the capability to rearrange anchors whilst preserving the irrelevance.  So under such circumstances, does the model's performance witness an enhancement?
2. The performance appears to be somewhat sensitive to the fine-tuning of hyperparameters, such as on CALTE7. It would be valuable if the authors could delve deeper into the potential implications of anchor noise and provide additional insights to mitigate its influence.
3. A central concept in this paper involves leveraging self-expression to derive anchor proximity and thereby boost the clustering outcomes. However, how does this contribute to improving the anchor quality? Moreover, if constructing anchors through other means, whether this remains functional?
4. Although the model incorporates anchor relations, it appears to fall short in capturing deeper cross-view complementarities. When encountering the scenarios where anchor quality exhibits substantial variability across different views, how does the model perform? When integrating complementarities at the graph similarity level, how is the performance?

**Questions For Authors:**

Please see the weaknesses.

**Relation To Broader Scientific Literature:**

This paper constructs a multi-view clustering method that contains both anchor-sample and anchor-anchor proximity and is without involving spectrum and baseline-view. It considers anchor-anchor characteristics and directly outputs clustering results without variance.

**Theoretical Claims:**

The solving procedure is presented detailly.

---

> ### Author Rebuttal · Authors · 2025-03-31
>
> **Q1:** Can the binary be further refined? How is the performance under orthogonality?
>
> **A1:** Thanks. Orthogonal constraints (OC) could deteriorate semantic topological continuity and limit the model's expression ability. Moreover, they will change the value and distribution of anchors. The following experiments further illustrate the distinction. BE is the results based on binary elements.
>
> |Dataset|DERMATO|CALTE7|Cora|REU7200|Reuters|CIF10Tra4|FasMNI4V|
> |:---:|:---:|:---:|:---:|:---:|:---:|:---:|:---:|
> |ACC||||||||
> |OC|83.46|78.46|52.37|**26.31**|52.79|27.12|55.42|
> |BE|**85.47**|**80.66**|**52.44**|26.22|**54.26**|**26.83**|**57.36**|
> |NMI||||||||
> |OC|87.13|43.62|41.42|6.14|**31.91**|14.38|55.23|
> |BE|**89.97**|**45.25**|**43.70**|**6.25**|31.87|**15.64**|**59.21**|
> |Fscore||||||||
> |OC|83.26|73.59|40.16|26.97|42.78|18.94|50.67|
> |BE|**87.92**|**78.12**|**41.12**|**28.55**|**44.84**|**20.64**|**51.37**|
>
> BE outperforms OC in most cases, the reasons of which are that BE does not change anchor characteristics, and only rearrange them within the original space.
>
> **Q2:** Somewhat sensitive to hyperparameters. Exploring potential implications of anchor noise.
>
> **A2:** The parameter $\lambda$ is responsible for striking a balance between the reconstruction loss and the regularization of anchor self-expression, whereas $\beta$ modulates the degree of inter-view consistency. Inaccurate calibration may result in diminished performance or instability, particularly in the presence of noisy anchors.
>
> Anchor noise: (1) It could potentially compel the model to assimilate the noise patterns, precipitating an over-fitting condition; (2) It could potentially render the loss surface irregular, complicating the optimization process; (3) It could predispose the model to gravitate towards spurious correlations.
>
> Several potential strategies could be employed: (1) Implement a pre-filtering mechanism or confidence-based thresholds; (2) Develop a multi-anchor voting mechanism; (3) Integrate prior knowledge to dynamically adjust hyper-parameters; (4) Conduct dataset pre-processing to eliminate instances of anchor noise.
>
> **Q3:** How does self-expression improve the anchor quality? Still functional under anchors with other means?
>
> **A3:** Self-expression learning helps extract the geometric characteristics between anchors, and facilities the learning of anchors owing to the co-optimized mechanism. OSE-L and WSE-L are to validate the effectiveness of self-expression.  OSE-L and WSE-L are the results without/with self-expression under no learning.
>
> |||||||||
> |:---:|:---:|:---:|:---:|:---:|:---:|:---:|:---:|
> |ACC||||||||
> |OSE-L|60.42|43.88|27.96|14.33|25.32|19.73|41.27|
> |WSE-L|**65.64**|**64.59**|**30.24**|**16.68**|**27.20**|**24.08**|**47.21**|
> |NMI||||||||
> |OSE-L|64.37|35.68|5.88|1.01|1.38|12.57|44.79|
> |WSE-L|**69.84**|**37.95**|**33.54**|**1.06**|**1.43**|**12.98**|**47.07**|
> |Fscore||||||||
> |OSE-L|63.76|48.43|28.79|23.42|33.87|16.86|37.64|
> |WSE-L|**69.33**|**61.54**|**30.40**|**24.43**|**35.25**|**18.03**|**41.43**|
>
> The following WSE+L and WSE-L are to show the effectiveness of learning.'+L' denotes having learning.
>
> |||||||||
> |:---:|:---:|:---:|:---:|:---:|:---:|:---:|:---:|
> |ACC||||||||
> |OSE+L|71.51|49.05|30.35|16.75|47.05|26.69|52.15|
> |WSE+L|**85.47**|**80.66**|**52.44**|**26.22**|**54.26**|**26.83**|**57.36**|
> |NMI||||||||
> |OSE+L|83.97|40.21|6.02|2.53|23.19|15.48|58.13|
> |WSE+L|**89.97**|**45.25**|**43.70**|**6.25**|**31.87**|**15.64**|**59.21**|
> |Fscore||||||||
> |OSE+L|73.79|51.25|30.42 |28.54|43.04|17.70|46.77|
> |WSE+L|**87.92**|**78.12**|**41.12**|**28.55**|**44.84**|**20.64**|**51.37**|
>
> Besides, OSE-L and WSE-L show that under the anchors constructed by sampling, our mechanism is still functional.
>
> **Q4:** How does the model perform under varying anchor quality and graph level?
>
> **A4:** In the paper, we introduce a view-aware weighting to harmonize anchor significance on each  view.  OVA denotes the results without view-aware weighting.
>
> |||||||||
> |:---:|:---:|:---:|:---:|:---:|:---:|:---:|:---:|
> |ACC||||||||
> |OVA|84.59|76.73|44.94|16.82|47.84|25.26|52.69|
> |NMI||||||||
> |OVA|89.44|42.88|33.06|3.60|29.54|15.20|56.84|
> |Fscore||||||||
> |OVA|86.59|71.80|35.36|**28.77**|42.32|18.03|46.90|
>
> Perhaps, an instance-aware adaptive weighting paradigm  could be more advisable. Implementing anchor-wise importance calibration may yield more superior results  since anchors on the same one view also could exhibit diverse importance.
>
> About the performance at the graph similarity level, we first generate view-specific graph and then construct fusion (FGTC) to integrating complementarities .
>
> |||||||||
> |:---:|:---:|:---:|:---:|:---:|:---:|:---:|:---:|
> |ACC||||||||
> |FGTC|80.17|73.98|48.21|22.12|54.13|22.79|53.89|
> |NMI||||||||
> |FGTC|82.74|38.97|39.26|5.83|27.89|11.79|58.93|
> |Fscore||||||||
> |FGTC|80.87|70.46|33.72|21.63|42.87|18.84|47.73|
>
> Evidently, our model gathering information at the cluster-level receives better results in most cases.

---

### Official Review · Reviewer_ywBk · 2025-03-20

**Overall Recommendation:** 4

**Summary:**

The paper builds up double-track proximity for multi-view clustering to investigate the manifold structure among samples. In particular, it encodes anchor-anchor relation into anchor-sample similarity using self-expression learning and topology learning concurrently.  It relieves the restriction of baseline-view by assigning a matrix variable consisting of 0 and 1 to each view, and transforms anchors generated on each view via joint-learning to reshape them. Beyond these, a binary strategy is adopted to take shape the cluster indicator, and meanwhile the cluster indicator connects all views and the anchors on them.  A six-step optimization scheme with linear complexity effectively minimizes the loss function.

**Claims And Evidence:**

The main claims are that the anchor-sample plus anchor-anchor, alignment without reference, and no-spectrum contribute to producing more distinctive clusters. They are reflected by the comparative experimental results in Table 1 and multiple ablation results respectively. The running time comparison Figure 2 supports its linear time complexity.

**Essential References Not Discussed:**

No

**Experimental Designs Or Analyses:**

The designs are valid, and provide evidence for verifying the effectiveness of the presented method, together with thorough analyses.

**Methods And Evaluation Criteria:**

The proposed method displays the effectiveness in dealing with MVC problems, and demonstrates advantages against 17 clustering methods.

**Other Comments Or Suggestions:**

**Grammars:**

-  In line 198, ‘This spectrum-free model directly output ….’---> outputs;

-  In line 200, ‘$\boldsymbol{\alpha}$ play…’---> plays;

**Other Strengths And Weaknesses:**

-  The overall structure of this paper is cohesive, enabling readers to easily trace the progression.

-   Extensive comparing experiments, detailed discussions, and rounded ablations reinforce the contributions.


**Weaknesses:**

-  The demand about the parameter searching may undermine the practical capability.  There are two parameters requiring manual tuning. Moreover, according to the guideline in Section Q, their suggested ranges are also different. This may bring inadaptability for the model when encountering certain scenes with vague clusters.


-  The permutation learning procedure necessitates additional methodological clarification, as the determination of $\mathbf{T}_p$ relies on  traversal exploration according to the feasible region. Hence, the geometric properties of feasible region should be systematically characterized.

-  The baseline-view-free approach achieves  enhanced performance metrics compared to conventional approaches. Although empirical evidence currently supports this observation, a fundamental exploration of underlying mechanisms is warranted to strengthen the validity of this advantage.

**Questions For Authors:**

1. About the complexity, combined with the theoretical expressions, it seems that the computing of cluster indicators takes more time consumption. Does it affect the overall efficiency?  Is there any other way to mitigate this factor?

2. What is the reason for no-spectrum strategy spending less time consumption than traditional scheme?

3. In conjunction with the statement ‘one can project all anchors into a common space to make them have the same dimension’, by projecting raw data, we will own the anchors with the same dimension. Does the spatial misalignment emerge between anchors in this case? Furthermore, would this yield enhanced performance compared to the proposed DTP-SF-BVF method?

**Relation To Broader Scientific Literature:**

The idea of embedding anchor-anchor relation into anchor-sample similarity using point-to-point guidance after reshaping is worthy of recommendation in multi-view clustering literature.

**Theoretical Claims:**

Yes

---

> ### Author Rebuttal · Authors · 2025-03-31
>
> **Q1:** The parameter searching may undermine the practical capability.
>
> **A1:** Thanks! $\lambda$ governs the trade-off  between error reconstruction and anchor self-expression, while $\beta$ adjusts the cross-view consistency guidance. They collaboratively modulate the model's capacity. Despite the different searching ranges, our model, as evidenced by the results in Table 1, with the suggested ranges, achieves more competitive data clustering under multiple datasets with diverse scales, which reveals that our model is suitable for various scenes.
>
> The parameter searching indeed compromises practical applicability to some extent,  particularly in resource-constrained scenarios. Developing a dynamic parameter derivation mechanism based on data characteristics will be a compelling research direction, and we will conduct systematic investigations in future work.
>
> **Q2:** The geometric properties of feasible region about permutation should be systematically characterized.
>
> **A2:** Its feasible region consists of discrete elements, 0 and 1. There is only one 1-element in every column and row. These guarantee that the permutation only rearranges anchors in their original space, and does not alter the values of learned anchors.
>
> Besides, its objective is $\operatorname{Tr}\left(\mathbf{T} _p^\top \mathbf{G} _p \mathbf{T} _p \hat{\mathbf{A}} _p + \mathbf{T} _p^{\top} \hat{\mathbf{B}} _p\right)$ where
> $\hat{\mathbf{A}} _p=\lambda\mathbf{H} _{p}+\boldsymbol{\alpha} _p^2\mathbf{M} _p-2\lambda\mathbf{S} _{p}^{\top}$ and$\hat{\mathbf{B}} _p=-2\boldsymbol{\alpha} _p^2 \mathbf{J} _{p}$. It exhibits quadratic operation about permutation elements and inherent non-separability, which causes difficulties in optimizing. Note that the permutation is with the size of $m\times m$, we can utilize one-hot vectors to constitute it.
>
> **Q3:** A fundamental exploration for baseline-view-free mechanism.
>
> **A3:** This approach rearranges anchors without requiring any baseline views. Given that the misalignment arises from the differing order of anchors, we link each view to a learnable permutation, which allows for the flexible transformation of anchors within the original space. Moreover, it is compatible with anchors in an unified framework, and collaborates with their learning process. Selecting a baseline view introduces complicated solving procedure. An improperly chosen baseline view will also lead to inaccurate graph structure fusion. In contrast, we do not rely on any baseline views and automatically rearranges anchors. Besides, it is also proven to be with linear complexity, and hence does not harm the overall efficiency.
>
> **Q4:** Do the cluster indicators affect the overall efficiency? How to mitigate these factors?
>
> **A4**: Combined with Eq.(13), it mainly involves $\mathbf{W}$ and $\mathbf{Z}$. So, updating the cluster indicators will take $\mathcal{O}(d _pmk+d _pk^2+km^2+k^2m+nd _pm+nm^2+nmk)$ cost. Generally, $m$ and $k$ are largely smaller than $n$. $d_p$ is a constant and irrelevant to $n$. As a result, the cost on cluster indicators is linear to $n$. So, it does not affect the overall efficiency.
>
> Besides, the coefficients related to $n$ are $m$, $k$ and $d_p$, while $k$ is an inherent characteristic of datasets, and $m$ is set equal to $k$ in experiments. So, we can decrease the feature dimension to further improve the efficiency.
>
> **Q5:** What is the reason for no-spectrum strategy spending less time than traditional scheme?
>
> **A5:** The spectrum generation process and subsequent embedding grouping result in extra computational expense.
> Besides, we utilize binary learning to directly formulate cluster labels. This only involves the element comparing between two small-sized matrices $\mathbf{W}$ and $\mathbf{Z}$.
>
> **Q6:** How is the performance under anchors with the same dimension.
>
> **A6:** Thanks! Via projection, we indeed can make all anchors have the same dimensions (SD). However, even with the same dimensions, due to anchors still being  generated on each individual view, the anchor order across views could remain inconsistent. So, the spatial misalignment still exists. The following table summaries the comparisons. DD is the results under anchors with original diverse dimensions.
>
> |Dataset|DERMATO|CALTE7|Cora|REU7200|Reuters|CIF10Tra4|FasMNI4V|
> |:---:|:---:|:---:|:---:|:---:|:---:|:---:|:---:|
> |ACC||||||||
> |SD|74.51|65.89|47.43|24.13|52.84|23.14|54.92|
> |DD|**85.47**|**80.66**|**52.44**|**26.22**|**54.26**|**26.83**|**57.36**|
> |NMI||||||||
> |SD|76.82|37.64|38.62|5.24|28.61|12.73|56.94|
> |DD|**89.97**|**45.25**|**43.70**|**6.25**|**31.87**|**15.64**|**59.21**|
> |Fscore||||||||
> |SD|75.34|66.82|35.12|25.63|41.63|18.83|45.62|
> |DD|**87.92**|**78.12**|**41.12**|**28.55**|**44.84**|**20.64**|**51.37**|
>
> The paradigm using anchors with SD produces inferior results, the reasons of which could be that the projecting operation causes information loss and degenerates multi-view diversity, weakening the performance.

---

### Decision · Program_Chairs · 2025-05-01

**Decision:**

Accept (poster)

**Comment:**

All reviewers give positive scores. The idea is well-motivated. The overall structure and experiments of this paper are sufficient. The review of existing literature is comprehensive.